# Contrasting pathways to tree longevity in gymnosperms and angiosperms

Tree longevity is thought to increase in growth-limiting, adverse environments, but a quantitative assessment of drivers of global variation in tree longevity is lacking. We assemble a global database of maximum longevity for 739 tree species and analyse associations between longevity and climate, soil, and species' functional traits. Our results show two primary pathways towards long lifespans. The first is slow growth in resource-limited environments, consistent with the "adversity begets longevity" paradigm. The second pathway is through relief from abiotic constraints in productive environments. Despite notable exceptions, long-lived gymnosperms tend to follow the first path through slow growth in cold environments, whereas long-lived angiosperms tend to follow the second ("productivity") path reaching maximum longevity generally in humid environments. For angiosperms, we identify two mechanisms for increased longevity under humid conditions. First, higher water availability increases species' maximum tree height which is associated with greater longevities. Secondly, greater water availability increases stand density and inter-tree competition, limiting growth which may increase tree lifespan. The documented differences between gymnosperm and angiosperm longevity are likely rooted in intrinsic differences in hydraulic architecture that provide fitness advantages for gymnosperms under high abiotic stress, and for angiosperms under increased productivity or competition.

Trees are widely revered as the longest-living group of organisms on Earth, including some species that can live for thousands of years[1]. Their lifespans control the time that carbon resides in tree stems, the main carbon reservoir of trees, and are thus closely related to the total carbon storage of forests[2]. Tree lifespan is also a key ecological and demographic trait that defines a species' life history strategy, including its position along the fast-slow growth spectrum[3–5], and its sensitivity to environmental variability[6,7]. Thus, to understand forest dynamics, and its response to global change, it is imperative to shed light on the drivers of tree lifespan among species, and across the globe. These insights can support predictions of future responses to global change, for example, by providing Earth System Models with more realistic estimates of tree turnover[8,9]. Yet, we lack systematic analyses of the environmental and climatic drivers, and species' functional traits, associated with tree longevity.

Recent studies have provided strong evidence for increasing longevity towards high latitudes and altitudes[10,11], confirming early observations that some of the longest-lived conifer trees such as 5000 year old bristlecone pines and 2000 year old junipers are usually found near tree lines, and at the coldest and/or driest places. These observations led early dendrochronologists to conclude that "*adversity begets longevity*"[12]. However, multi-millennia old trees are not just found in cold environments with short growing seasons, or sites with limiting water availability. They also grow in some of the most productive and humid temperate forests (e.g., *Sequoia*, *Sequoiadendron*), in flooded swamp forests (e.g., *Taxodium*[13]), and in African savannas (e.g., *Adansonia*[14]). These contrasting observations demonstrate that our understanding of the global environmental controls on tree longevity remains limited.

In addition, large differences exist among species growing in the same community. In tropical forests, longevity varies from a few

✉ e-mail: r.brienen@leeds.ac.uk

decades for pioneers[15] to several centuries for other species[16,17]. These differences arise from the diversification of species along different axes of demographic variation. One of these axes is the well-established fast-slow continuum of life-history strategies shaped by trade-offs between growth and survival[18]. Greater investment towards growth and resource acquisition traits are expected to result in lower tree lifespans[3,19]. Consistent with this trade-off, previous analyses have shown that faster growth is negatively associated with species' longevity at the global scale[11,20]. However, very few studies have directly related species' longevity to their traits. Most investigations instead have focused on the association between traits and demographic rates, such as tree mortality[21,22]. For example, wood density and tree mortality have repeatedly been shown to be negatively correlated[21,23,24], but studies on relationships with longevity are less common[25]. In addition, mortality rates are far from uniform across a tree's life history[26], and mean mortality rates thus need not be correlated with maximum longevity across species and sites. Consequently, it remains unclear to what degree functional traits can be used to predict species' genotypic or phenotypic variation in longevity, and a systematic quantitative analysis on the roles of environment, climate, and plant functional traits on species longevity is needed to shed light on the controls on growth-longevity trade-offs. Such analyses will also enhance our understanding of how climate change may affect tree longevity, and thus forest dynamics. Whether environmental drivers associated with longer lifespans among species (i.e. related to their adaptive traits) – and the focus of this study - also drive longer lifespans within species across habitats (i.e. related to their phenotypic responses) is poorly understood but is beyond the scope of the present work.

Here, we compile a new database of tree longevity or maximum life expectancy of 739 tree species. This dataset is based primarily on tree ring records complemented with radiocarbon dating, historical age records and growth projections from short-lived species, and tree species from under-represented biomes to capture the full spectrum of life history strategies. While conifers from cold sites are well represented (Supplementary Fig. 1b), our new dataset contains many tropical and short-lived (mostly broadleaved) species not included in previous analyses[11,27,28]. We use this database to assess the association of species' longevity with global environmental gradients and functional traits, and aim to answer the following questions: (1) How is longevity associated with temperature, precipitation, and soil characteristics? (2) Can species' traits predict variation in species' longevity? And, (3) What are the relative roles of environment, traits and species growth rates on tree longevity? Previous studies showed that angiosperms and gymnosperms differ strongly in their growth strategies[29], photosynthetic capacity[30], hydraulic traits[31–33] and longevity[1,10,11,34], which we confirm in our analysis of the full dataset. Given that these taxonomic groups diverged over 300 million years ago[35] and that they differ in their global distribution[36], we primarily analyse the results for these groups separately to avoid any phylogenetic bias in the results. This represents the first effort aimed at disentangling the effects of climate, environment and species' functional traits on tree species longevity.

## Results
### General patterns of variation in tree longevity
The oldest trees are typically found in mountain ranges at around 40° latitude such as the American Cordillera, Himalayas and Mediterranean mountains. However, trees over 1000 years old can also be found in tropical and subtropical regions (Supplementary Fig. 1a). We found that gymnosperm species live significantly longer (mean = 717 years, s.e. = 47 years) than angiosperm species (mean = 296, se = 12 years), and that there is a strong negative relationship between growth rate and longevity (Fig. 1a). When analysed by biome, we found that longevity increased from warm to cold climates (i.e., tropical <mediterranean <temperate mixed <boreal <temperate conifer biomes), and within (sub)tropical biomes from drier to wetter climates (dry grasslands and savannas & dry forest <moist forests, Fig. 1b). Species from tropical dry ecosystems (i.e., dry broadleaved forests, and grasslands, savannas and shrublands) had the lowest

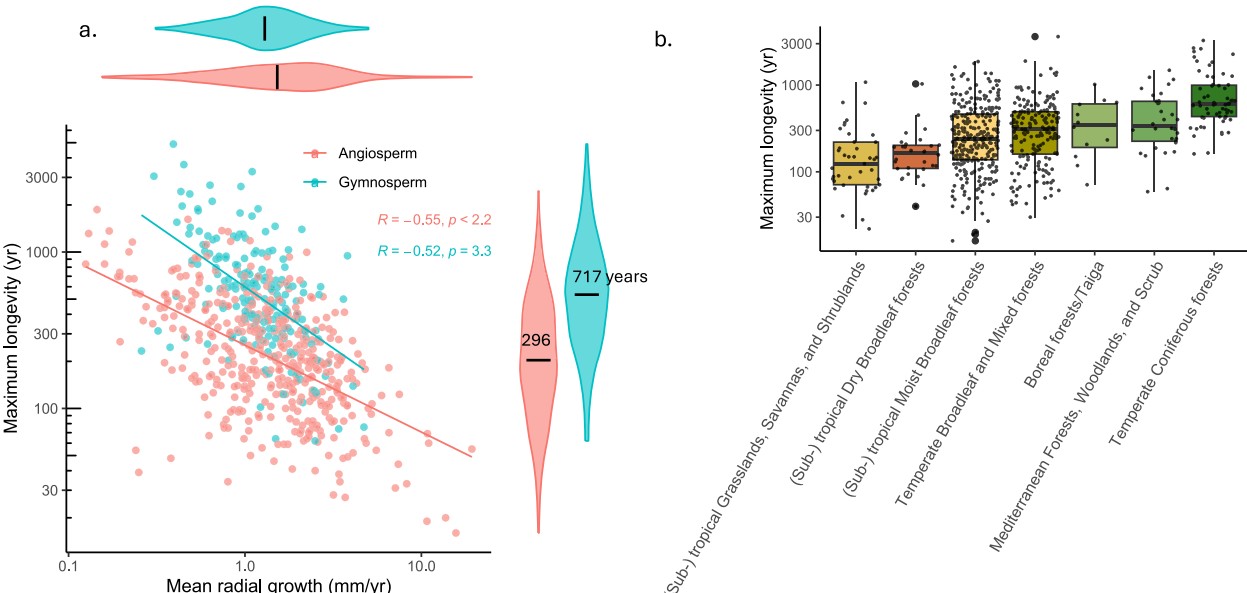

**Fig. 1 | Effects of growth and biome on species' longevity.** Relationship between species' radial growth and tree longevity (**a**) and variation in tree longevity across biomes (**b**). Lines in the violin plots (**a**) and the box plots (**b**) show the median of the groups or biomes. The median and mean tree longevity was 204 years and 296 years (s.e. = 12 years) for angiosperms, and 534 and 717 (s.e. = 47 years) for gymnosperms. Both radial growth and maximum longevity were log₁₀-transformed, and the lines in panel a corresponds to a linear regression. Box plots display the median (central line), upper and lower quartiles (box edges), and whiskers that extend to values within 1.5× interquartile range (IQR). Larger points beyond the whiskers are outliers.

longevity, whereas coniferous temperate forest species displayed the highest longevity.

## Covariation of species' longevity with climate and soil

The effects of climate on tree longevity, at the global scale, differed between angiosperm and gymnosperm species (Fig. 2, Supplementary Tables 1, 2). Temperature had a greater influence on gymnosperm longevity (Fig. 2a), while precipitation had a stronger effect on angiosperm longevity (Fig. 2b). Longevity of gymnosperm species increased towards colder climates (Fig. 2a) and for species growing closer to the sub-arctic and alpine tree lines (Fig. 2c). The effect of mean annual temperature on tree longevity is in almost equal measures due to decreased growing season temperature and a shortening of the growing season length in colder climates (see Supplementary Table 2). Angiosperm longevity increased with precipitation metrics and climate moisture index (CMI) that takes potential evapotranspiration into account (Supplementary Table 1). The strongest correlation was found with precipitation during the driest quarter of the year (Fig. 2b). These climate relationships are robust when excluding species with lower confidence in their longevity estimates (Supplementary Fig. 3, Supplementary Table 3c). We further find that gymnosperms show a decrease in longevity with increasing site productivity, whereas angiosperms show a weak positive relationship with site potential net primary productivity (Fig. 2d).

Soil cation exchange capacity (CEC) and soil pH showed bivariate relationships with tree longevity in both groups (Supplementary Tables 1, 2). However, these effects on longevity are confounded by covariation with climate at the global scale. For example, in the angiosperm dataset, soil pH strongly covaried with precipitation and its effect on longevity disappeared when controlling for the effects of precipitation, temperature and CEC. Similarly, we found that a negative global relationship between CEC and temperature partially obscured a negative, but weak effect of temperature on longevity in angiosperms (Supplementary Table 2). That is, increased longevity due to colder temperatures at higher latitudes was partially offset by decreased longevity due to greater CEC at higher latitudes. For gymnosperms, we find no direct effect of CEC or soil pH after accounting for the effects of temperature and precipitation (Supplementary Table 2).

## Covariation of traits and tree longevity

Many of the studied traits showed significant bivariate relationships with longevity across the full dataset, including both angiosperms and gymnosperms (Supplementary Table 1). However, most of these relationships disappeared when the data were analysed separately for angiosperms and gymnosperms. Of all the studied traits, the only traits that remained strong predictors for longevity within gymnosperms and angiosperms, were species' radial growth (Fig. 1a) and species' maximum tree height (Fig. 2f). For gymnosperms, no other functional traits were significantly related to longevity. For angiosperms we found significant, but weak, effects of leaf nitrogen (Fig. 1h) and wood density (Fig. 1e). Our results remained largely robust even when excluding growth projections from our analysis although the relationship with wood density for angiosperms disappeared (Supplementary Fig. 4). Within the two groups, we found no effects of hydraulic traits (P50, hydraulic safety margin-HSM, conduit size, and density, Supplementary Table 1).

As plant trait variation at the global scale was influenced by climate, we used Structural Equation Modelling (SEM) to disentangle the direct and indirect effects of climate and traits on species' longevity. We included in this analysis key climate variables, i.e., temperature and precipitation during the driest quarter of the year, and those traits that showed associations with longevity, i.e., radial growth, maximum tree height and wood density. Results of the SEM are shown in Fig. 3 with detailed statistics in Table 1. About two third of the effect of

precipitation on longevity in angiosperms was due to its influence on tree height and growth, while one third was due to the direct effect of precipitation (Table 1). Higher precipitation was associated with greater maximum tree height and reduced tree radial growth, both contributing to increasing longevity in angiosperms (Fig. 3a, Table 1). Temperature had a modest direct negative effect on longevity, while indirect effects of temperature via wood density and radial growth cancelled each other out. For gymnosperms, all of the effect of temperature was due to its influence on growth (i.e., increased tree growth at warmer sites reducing longevity) while our analysis did not detect a direct effect of temperature (Fig. 3b, Table 1). We find no significant effect of precipitation on gymnosperm longevity.

## Association with the global plant trait spectrum

To overcome the limitation of the limited amount of trait data available for each species, we also perform the analyses with a more complete set of imputed trait data from ref. 37. Despite the additional uncertainty of these infilling approaches, this approach provides a more complete perspective on the association of longevity with a wider range of traits commonly used in global plant trait analyses. These data showed that for angiosperms longevity was associated with plant economics traits such as wood density, specific leaf area (SLA), and leaf nitrogen, and with the orthogonal axis of plant stature traits including plant height, seed mass, and conduit density (Supplementary Fig. 8). In contrast, in gymnosperms, there was no association with the plant economic traits (SLA and wood density) and the association with tree height was contrary to that expected (i.e., greater tree height resulting in lower longevity).

## Evaluation of phylogenetic effects

We find that both groups show significant phylogenetic signals in longevity variation (Pagel's $\lambda$ for Gymnosperms = 0.905, $p < 0.001$; Pagel's $\lambda$ for Angiosperms: 0.46, $p < 0.001$). We therefore checked whether our results are robust with regard to these phylogenetic relationships. To this end, we first repeated the correlation analysis for key variables shown in Fig. 2 whilst accounting for phylogenetic relationships between species and find that all remain significant with very little change in the correlation strength (see Table 2). The only correlations that switched to non-significant are those between angiosperm longevity and wood density ($p = 0.069$) and potential site productivity ($p = 0.108$). Secondly, we reanalysed the data using linear mixed effects models incorporating family as a random effect. This proves that relationships with climate hold up within families (i.e., standardized effect of temperature on gymnosperm longevity = −0.157, $t$-value = −7.23, $p < 0.001$, $n = 205$, standardized effect of precipitation during the driest quarter on angiosperm longevity = +0.107, $t$-value = 6.43, $p < 0.001$, $n = 534$). Finally, we reanalysed the results using genus level means of longevity and climate. It confirms again that longevity-climate relationships remain robust within both groups (Supplementary Fig. 8a, b).

## Discussion

Our analyses of a greatly expanded and diverse dataset confirmed that gymnosperms live longer than angiosperms (mean = 744 vs. 296 years). This difference is larger than that reported in previous datasets[1,10,11,34], likely because of the addition of a larger number of short-lived, angiosperm species. We also confirmed the existence of a strong negative relationship between radial growth and longevity (Fig. 1a), consistent with life history theory, which postulates that trees face allocation trade-offs[3,4,18]. We focus our discussion below on the new aspects of our study which includes 1) clear differences between angiosperms and gymnosperms in the association of longevity with climate independent of phylogeny, 2) insights in the ecological factors contributing to greater longevity for gymnosperms in cold climates (i.e., low growth) and for angiosperms in humid climates (i.e.,

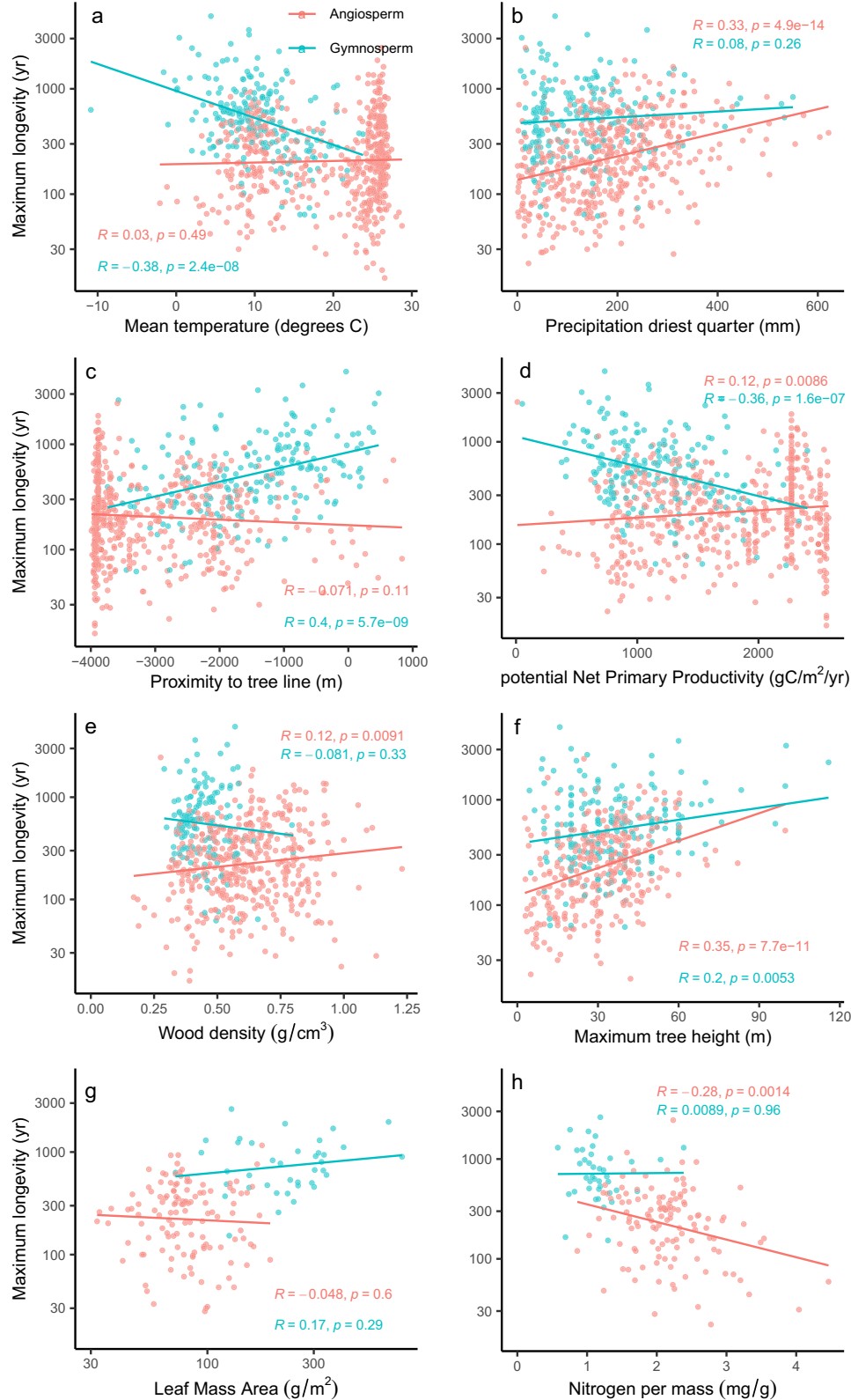

**Fig. 2 | Relationships between species' longevity and most important climate, environmental and trait variables.** Variables include species' mean annual temperature (**a**) and precipitation during the driest quarter (**b**), species' mean proximity to the estimated tree line position (**c**), potential net primary productivity (**d**), and species' functional traits including wood density (**e**), maximum species tree height (**f**), and leaf mass per area (**g**) and leaf nitrogen (**h**). Trend lines are fitted using a linear model. Statistics show the correlation coefficient (*R*) and significance levels (*p*) for linear regressions. Data were grouped by the two major taxonomic groups in the dataset, angiosperms and gymnosperms. Sample sizes are shown in Supplementary Table 1. Note that data for Leaf Mass per Area were $\log_{10}$ transformed. Phylogenetically controlled correlations using phylogenetic generalised linear models (PGLS, see methods) for all variables are shown in Table 2.

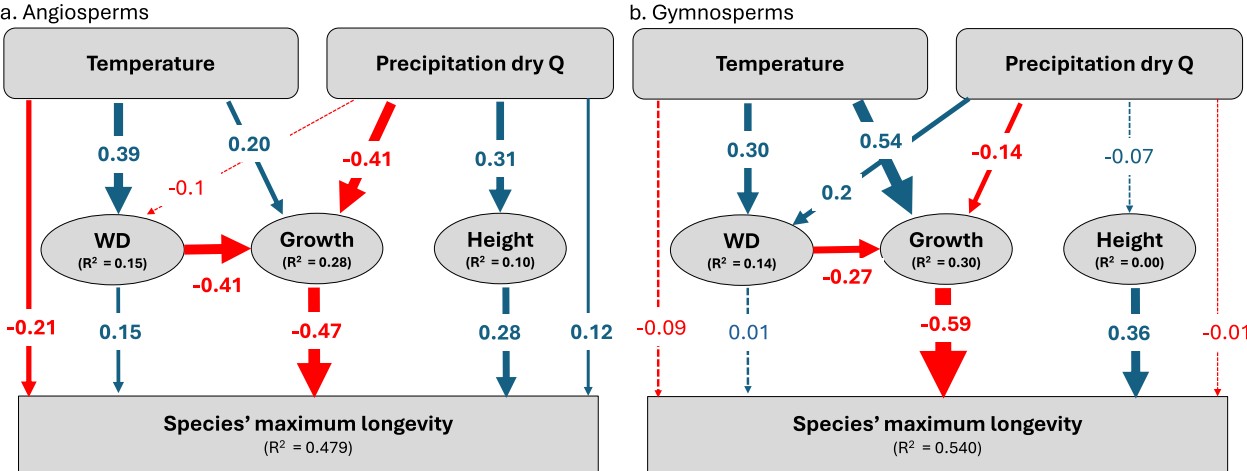

**Fig. 3 | Direct and indirect effects of climate and traits on species' longevity.** Structural equation modelling (SEM) of the multivariate effects of climate (mean annual temperature and precipitation of the driest quarter) and traits (wood density-WD, mean radial growth, and maximum tree height), on species' longevity for angiosperms (**a**) and gymnosperms (**b**). Solid lines represent significance influence ($p < 0.05$) with the numbers indicating the strength of the relationships. Blue indicates positive and red negative relationships. Model statistics and direct and indirect effects and sample sizes are shown in Table 1. Omission of non-significant variables and paths, resulted in a better fit for gymnosperms, but for comparison purposes we applied the same model to both groups.

**Table 1 | Statistics of the structural equation modelling (SEM) for the effects of climate and traits on species longevity for angiosperms and gymnosperms**

|  | Direct | Indirect | Total effect |
|---|---|---|---|
| **Angiosperms (n = 198 species)** | | | |
| Mean annual temperature | −0.212 (p < 0.001) | 0.041 (p = 0.360) | −0.172 (p = 0.006) |
| Precipitation of the driest quarter | 0.123 (p = 0.040) | 0.245 (p < 0.001) | 0.368 (p < 0.001) |
| *Precipitation ⇒ Growth ⇒ Longevity* | | 0.192 (p < 0.001) | |
| *Precipitation ⇒ Max Tree height ⇒ Longevity* | | 0.086 (p < 0.001) | |
| Wood density | | | 0.152 (p = 0.013) |
| Mean radial growth | | | −0.472 (p < 0.001) |
| Maximum tree height | | | 0.278 (p < 0.001) |
| Model statistics[a]: $\chi^2 = 5.50$, $p = 0.139$, CFI = 0.989, SRMR = 0.036, RMSEA = 0.065. | | | |
| **Gymnosperms (n = 128 species)** | | | |
| Mean annual temperature | −0.088 (p = 0.240) | −0.284 (p < 0.001) | −0.373 (p < 0.001) |
| *Temperature ⇒ Growth ⇒ Longevity* | | −0.282 (p < 0.001) | |
| Precipitation of the driest quarter | −0.006 (p = 0.919) | 0.090 (p = 0.123) | 0.083 (p = 0.308) |
| Wood density | | | 0.002 (p = 0.978) |
| Mean radial growth | | | −0.593 (p < 0.001) |
| Maximum tree height | | | 0.362 (p < 0.001) |
| Model statistics[a]: $\chi^2 = 13.82$, $p = 0.000$, CFI = 0.929, SRMR = 0.083, RMSEA = 0.168. | | | |

Parameters are reported as standardized effects.

Species longevity and mean radial growth were $\log_{10}$-transformed. Significance values for the effects are indicated in brackets.

[a]*CFI* Comparative Fit Index, *SRMR* standardized root mean square residual, *RMSEA* root mean square error of approximation.

increased tree height and competition), and 3) the existence of relatively weak effects of functional traits on tree longevity, once phylogenetic differences between angiosperms and gymnosperms are taken into account.

## Association of tree longevity with climate

We found marked differences in climate-longevity relationships between gymnosperms and angiosperms. While gymnosperm longevity declines with increasing temperature, angiosperm longevity declines with increasing aridity. Previous research identified similar effects of temperature and precipitation on longevity when gymnosperm and angiosperm data were pooled[10,11]. Here, a database with greater replication allowed us to show that climate-longevity relationships differ between gymnosperms and angiosperms.

The negative effect of temperature on gymnosperm longevity has been known for a long time[12], but the underlying cause of this relationship remained unclear. Our analysis suggests that all of the temperature effect is indirect due to the effect of temperature on radial growth (Fig. 3, Table 1). It is well known that in ecozones noted for their low temperatures and typically short growing season length[38] and low nutrient supply rate[39], conifers tend to have a constrained rate and duration of cambial cell division and expansion[40,41], low leaf photosynthesis[42], and low needle nitrogen[43,44], as well as long needle lifespan[43,44] (the latter two of which are known to co-vary with photosynthesis and growth rate)[43,44]. This, in turn, could prolong a trees' lifespan, as longevity declines exponentially with radial growth (Fig. 1a). The simultaneous decrease in longevity and increase in growth with temperature is also consistent with the suggested control

**Table 2 | Standard and phylogenetically controlled correlations for the key variables affecting variation in longevity (cf. Supplementary Table 1)**

| | Angiosperms | | | | | | Gymnosperms | | | | | |
|---|---|---|---|---|---|---|---|---|---|---|---|---|
| | Standard correlation | | | Phylogenetically controlled | | | Standard correlation | | | Phylogenetically controlled | | |
| | *R* | *p* value | *n* | *R* | *p* value | *n* | *R* | *p* value | *n* | *R* | *p* value | *n* |
| Mean radial growth[a] | **−0.564** | **0.000** | **429** | **−0.507** | **0.000** | **381** | **−0.567** | **0.000** | **184** | **−0.467** | **0.000** | **179** |
| Mean annual temperature | 0.030 | 0.488 | 534 | 0.038 | 0.405 | 476 | **−0.378** | **0.000** | **205** | **−0.420** | **0.000** | **200** |
| Precipitation driest quarter[b] | **0.332** | **0.000** | **488** | **0.277** | **0.000** | **432** | 0.080 | 0.258 | 203 | 0.072 | 0.312 | 198 |
| Proximity to tree line | −0.071 | 0.108 | 519 | 0.048 | 0.302 | 462 | **0.397** | **0.000** | **201** | **0.429** | **0.000** | **196** |
| Potential Net Primary Productivity | **0.115** | **0.009** | **517** | 0.075 | 0.108 | 460 | **−0.359** | **0.000** | **201** | **0.423** | **0.000** | **196** |
| Wood density | **0.124** | **0.009** | **441** | 0.091 | 0.069 | 399 | −0.081 | 0.333 | 144 | 0.123 | 0.146 | 140 |
| Maximum tree height | **0.351** | **0.000** | **325** | **0.294** | **0.000** | **291** | **0.200** | **0.005** | **198** | **0.261** | **0.000** | **193** |
| Leaf Mass per Area[a] | −0.048 | 0.599 | 124 | 0.082 | 0.392 | 109 | 0.175 | 0.287 | 39 | 0.282 | 0.082 | 38 |
| Nitrogen per Mass | **−0.284** | **0.001** | **124** | **0.220** | **0.020** | **109** | 0.009 | 0.957 | 40 | 0.002 | 0.991 | 39 |

Standard correlations are Pearson's correlations and phylogenetic controlled correlations were performed using phylogenetic generalised linear models (PGLS) (see Methods).
Statistical significance levels are based on uncorrected *p*-values (i.e., no Bonferroni correction for multiple comparisons).
[a]These data were log$_{10}$-transformed for the analysis.
[b]Correlations with precipitation and Climate moisture index exclude species that have their main occurrence in floodplains or wetlands.
Bold numbers represent significant correlations.

of metabolism and oxidative processes on longevity of tree stems (i.e., the rate of living (ROL)) theory of aging[45], as metabolism is strongly controlled by temperature.

While there is an overall tendency for gymnosperm longevity to increase towards cold sites, we found that a subgroup of gymnosperms reached their optimum lifespans in productive, dense forest sites where growth is not restricted by abiotic limitations. These include large conifer species from taxa within the Araucariaceae, Podocarpaceae, Cupressaceae and some Pinaceae. The most famous of these are the redwoods of California (*Sequoia, Sequoiadendron*), Patagonian Alerce (*Fitzroya*), bald cypresses (*Taxodium*) and kauri (*Agathis*). A targeted analysis of this group revealed that variation in longevity for these gymnosperms is associated with maximum tree height and is not related to temperature or growth, in contrast to most other gymnosperms (Supplementary Fig. 6). This group instead seems to adopt a strategy of achieving old ages through sustained growth enabled by ample water availability and relatively large leaf area, which allows trees to reach large statures and outcompete other trees[46]. Additional features that allow these species to achieve old ages are the production of wood that is resistant to beetle and fungal infections[47] and high fire resistance due to bark that can exceed 15 cm in *Sequoia* and 30 cm in *Sequoiadendron*[48].

The effect of temperature on longevity in angiosperms is much weaker than for gymnosperms, with negative relationships emerging only after the influence of traits and other environmental factors are accounted for (Fig. 3, Table 1). An unexpected result of our study was a positive effect of precipitation on longevity in angiosperms (Figs. 2b, and 3). These results are not in line with the general idea that "adversity begets longevity", and previous analyses have produced conflicting outcomes, finding both positive[10,11] and negative effects[27,34] of precipitation on longevity. While some of these discrepancies may be attributed to differences in datasets and their geographic extent (e.g., tree ring data only[27,34] vs. more comprehensive datasets), or analytical approach (e.g., site-level[27,34] vs. species-level analysis[11]), the positive association between longevity and precipitation across the largest global dataset of angiosperm species is clear. By accounting for indirect effects of precipitation on longevity through tree growth and traits, we show that the majority of the effect of precipitation is explained by the effects of precipitation on tree height and growth (Fig. 3). Increases in precipitation lead to greater maximum tree height[49–52], which in turn is associated with tree longevity (Fig. 3). In addition, we found a weak indirect effect of precipitation on longevity through the effect of precipitation on growth (i.e., a negative

relationship between radial growth and precipitation, see Supplementary Fig. 5b, Fig. 3). While this seems counterintuitive, it most likely reflects increases in competition for light in more productive, wetter sites at lower latitudes, where trees form taller and denser canopies and standing biomass is larger[53,54]. This increases competition for light, and restricts growth of trees in the understory[55,56]. Tree ring studies on both temperate and tropical angiosperms indeed show that trees can remain suppressed and survive in the understory for decades to centuries[57–59], and radiocarbon and other dating techniques suggest that some trees may even reside in the understory of tropical moist forests for several centuries[17,60]. Trees in shaded environments have lower growth due to low light but are able to persist in such conditions due to both phenotypic and genotypic energy-saving shifts towards low carbon cost foliage with low respiration rates and often extended leaf life-spans[61,62]. These may represent important ecological processes that contribute to extending longevity of shade-tolerant species[1]. Finally, increasing precipitation could also prolong tree longevity directly by reducing the risk of drought-induced embolism, an important mortality mechanism in trees[63]. However, despite differences between angiosperms and gymnosperms, we do not detect strong relationships between longevity and hydraulic traits within these groups (P50, HSM, conduit size or density, Supplementary Table 1). This may point to a convergence of species' hydraulic safety margins across the globe[64], although other studies report that variation in hydraulic safety margins are a good predictor for drought related mortality[63,65]. More data on hydraulic properties are needed to fully explore these effects for a greater number of species.

### Why do gymnosperms and angiosperms differ in their climate-longevity relationships?

The most likely cause for diverging climate-longevity relationships between angiosperms and gymnosperms is the intrinsic and phylogenetically determined differences in their hydraulic architecture, hydraulic safety margins, and ability to repair xylem embolisms, resulting in differences in resistance to climate stress[32,64,66] and competitiveness[29,67]. Gymnosperm tracheids are, on average, narrower and shorter than angiosperms' multicellular vessel conduits, which makes them less vulnerable to cavitation and embolism during drought or freeze-thaw events[64,68]. Gymnosperm tracheids also have bordered pits that provide greater embolism resistance[33,69] which may allow gymnosperms to survive in harsh environments such as at cold sites and some semi-arid regions[29,70].

Angiosperms are generally more efficient than gymnosperms at transporting water and have higher photosynthetic capacity[30,67], making them more competitive in warm, wet, resource rich environments[29,36]. However, observed trade-offs between hydraulic conductance and safety from embolism in angiosperms limits height attainment, and thus tree lifespan, for this group in dry environments, whereas such trade-offs are weaker in gymnosperms[51]. These mechanisms may help explain why gymnosperm longevity was unaffected by precipitation, and why longevity responses to climate differ between gymnosperms and angiosperms. However, we acknowledge that our analysis is correlative and that climate explains only a modest amount of variation in longevity. Additional features that may explain some of the differences in response to climate, include a greater ability for gymnosperms to compartmentalise attacks by diseases, insects, and fungi[25,71], and their greater allocation to defensive compounds and lignin concentrations[72]. Finally, differences in the climate response may also emerge from environmental filtering for different climates, as gymnosperms occur predominantly in environments that experience freezing during some parts of the year. In line with this we find that separation of the dataset into species occurring in freezing versus non-freezing environments results in similar relationships with temperature and precipitation as those observed for gymnosperms and angiosperms (Supplementary Fig. 9). These results show that more research is needed to assess interactions between environment and phylogeny, and to fully disentangle the relative roles of these various mechanisms in controlling tree survival and explain differences in associations between longevity and climate. Further investigations could also assess trait-longevity associations within ecosystems and biomes and explore to what degree longevity variations are related to external disturbances (e.g. fire, hurricanes).

## Can functional traits predict tree longevity?

Based on plant economic theory we expected that allocation in resources towards one purpose, e.g. survival and maintenance, comes at the cost of investment in other functions, e.g., resource acquisition and growth[73]. Across the full dataset, we indeed found evidence for such trade-offs with greater longevity for species with safer hydraulic systems (i.e., higher HSM, lower P50, smaller conduits and higher conduit density), and species with slower, less acquisitive leaf traits (i.e., higher LMA, lower photosynthetic assimilation and nitrogen per mass) (see Supplementary Table 1). Leaf nitrogen was the single strongest predictor of longevity across all species, among all climate and trait variables (Supplementary Table 1). However, these global trait longevity associations were almost entirely driven by phylogenetically determined trait differences between the angiosperms and gymnosperms, as separate analysis for these groups showed fewer and relatively weak relationships.

Within gymnosperms, none of the plant economics traits were associated with longevity, while within angiosperms only wood density and leaf nitrogen were significantly related to longevity (Supplementary Table 1). Effects of wood density were weak however (Fig. 2e). These results are somewhat surprising, as wood density provides trees with mechanical strength[74], protection from fungi and pathogens[72], and resistance to drought embolism[75], all of which affect tree survival, and were thus expected to exert strong control on tree longevity. The weak relationship between wood density and longevity is consistent with studies on tropical forests that found species' wood density to be a relatively poor predictor for tree mortality (i.e., $r^2 \sim 0.2\text{-}0.4$), while species' mean radial growth rate was a better predictor[23,24,76]. The only study that previously related longevity with wood traits showed that wood density was weakly related to longevity for North American angiosperms, but not for gymnosperms, and that volumetric heat content (i.e., energy released upon burning) was a better predictor for tree longevity[25]. Volumetric heat content combines both structural and chemical investments, highlighting the strong role that chemical

defence compounds may play in prolonging tree lifespan. These secondary compounds may be especially important for gymnosperms, as this group has slower wood decomposition rates than angiosperms despite a lower mean wood density[72], and lacks a relationship between wood density and longevity (Fig. 2e). Apart from wood density, the only other economics trait that showed a significant relationship with longevity was leaf nitrogen for angiosperms, but the relationship was weak.

In contrast to the relatively poor within-group predictive power of traits associated with the fast-slow plant economic trait spectrum, species maximum height was significantly related to species' longevity within both groups (Fig. 2f) and explained a larger, though still modest, share of variation. Maximum tree height integrates across many aspects of plant stature and is the second most important functional traits axis in trees, orthogonal to the fast-slow axis[37,77,78]. Large statured trees have greater exposure to wind gusts, which can increase the likelihood of stem breakage or windthrow[79–81] and increased exposure to lightning[82]. Yet, at the global scale, the greater likelihood of damage or mortality posed by these factors is outweighed by the advantages conferred by greater maximum height. Large-statured species may reach greater tree ages by greater investment in growth in the adult phases allowing them to outcompete smaller statured species for light and survive longer[21]. This guild of trees has been described as long-lived pioneers, and occur in both tropical[21] and temperate humid forests[83]. Examples of long-lived pioneers for temperate humid forests include the previously mentioned group of large conifer species (e.g., *Sequoia*, *Sequoiadendron*, *Fitzroya*, *Agathis*). Faster growth rates for some of these large, long-lived pioneers may also increase survival by reducing the exposure to some pathogens and herbivores, and by increasing bark thickness that provides protection from surface fires[48,84]. Interestingly, the longevity-height associations break down in those conifer species that usually grow in low competition sites where cold or dry conditions limits radial growth and where reproductive success does not depend on outcompeting other trees (Supplementary Fig. 6). Examples for this group are cold- and dry-adapted gymnosperms such as *Pinus longaeva* and several *Juniperus* species that reach old ages (>1000 yrs) with low maximum statures. These results are consistent with studies showing that the biggest trees are not always the oldest[1,58,85], and contrasts strongly with the height-longevity associations for angiosperms (Fig. 2f).

In summary, our analyses showed that most species-specific traits provided relatively weak relationships with species' tree longevity within both groups. This poor predictive power is not very different from the findings of other global studies on trait variation and species life history[22], and could be caused by various factors[86]. First, we used species' mean trait values while traits may vary strongly from site to site, or even within the site, for the same species[86]. Secondly, we mixed trait data from a large variety of ecosystems from across the globe, but traits are known to covary with global variation in soil and climate[37,74,77]. For most analyses, the use of species mean trait values should provide robust results due to limited species trait crossover across environments[87], but we cannot discount that this broad approach may have contributed to obscuring the relationships between traits and longevity at a global scale. Finally, only a small number of species in our data set had a larger set of directly measured traits, which limited the power of our analyses. To overcome this problem we had to rely partially on imputed trait data[37] to assess relationships with multiple traits at the same time (Supplementary Fig. 7). Future research should prioritize expanding databases to increase the coverage of the longevity and trait data and include additional traits relevant to longevity such as defensive compounds and bark properties.

While we could not fully assess trait-longevity associations for a larger range of traits, some differences emerged between angiosperms and gymnosperms. Angiosperm longevity was associated with the two dominant axes of trait variation, plant economics (i.e., fast-slow) and

plant size (small-big). In contrast, for gymnosperms, there was only a relationship with plant size for a small subgroup, and almost no association with plant economics traits. These differences may be related to the greater variation in hydraulic and leaf morphological traits in angiosperms compared to gymnosperms. Evolution of a more flexible hydraulic architecture[31,88] may have resulted in greater diversification of angiosperms along the fast-slow and small-large life history axes, allowing adaptation to different niches in relatively stable, but productive, competitive environments. This may have been further facilitated by faster demographic rates of angiosperms (i.e., more than two times shorter mean longevity), which itself may increase species diversification rates[89]. In contrast, gymnosperms predominantly seem to lack diversification along the fast-slow spectrum possibly because of their "slower" hydraulic systems that are more adapted to stress tolerances in low competition environments[31,33,64].

## Summary and outlook

We identified two fundamental pathways in which trees can become old. The first is slow growth driven by strong resource limitations, i.e. abiotic stress in the form of low temperature and/or poor soils, increasing longevity for a number of conifer species that confirms the long-established notion that "adversity begets longevity". The second path to old ages involves the opposite conditions, i.e., relief from abiotic constraints at wet and productive sites. This second path is primarily found in angiosperm species, but also in a smaller group of gymnosperms growing in productive forests. In this productivity pathway increases in competition limits individual growth for trees in the understory and allows others to reach large statures, both of which are associated with greater longevity.

Differences in longevity among species are ultimately the outcome of long evolutionary divergence between species with specific adaptations arising to provide the fitness advantages conveyed by longevity in these very different environments. These two pathways to old ages in trees represent two extremes along a competition spectrum; low resource and low competition at one end, high resource and high competition at the other end. Indeed, these strategies are generally located within two corners of Grime's "C-S-R" triangle[90]; old conifers at sites with adverse growing conditions (i.e., low temperature, short growing seasons, poor soils, etc.) would occupy the lower left-hand "S" corner representing extreme "stress tolerators" (low competition, extreme stress), whereas long-lived canopy and emergent trees in forests would occupy the upper "C" corner, representing the ultimate "competitors" (high competition, low stress) able to acquire an unequal share of the most limiting resource, light in the case of most productive forests.

## Methods

### Longevity database

We compiled a new extended database of tree species longevity, that includes a total of 739 species, 205 gymnosperms and 534 angiosperms. Species longevity was defined as the highest age of the above ground stem tissue recorded for a given species. While below ground tissues or genets may be older than our definition of stem longevity, we did not include these ages. As we focus here on variation in tree longevity, or tree maximum lifespan, between species, we ignored variation within species arising from differences in climate[58], soil[20] and disturbance frequency[91]. Age estimates were obtained from a mixture of tree ring records, radiocarbon dating, growth projections and historical records sourced from original databases and literature (see Supplementary Table 3a for key datasets). For most species (78%), the highest age estimates were based on tree ring counts. Where possible we recorded the mean radial growth increment for each species, calculated as the average ring width across all sites or obtained from reported mean ring width or diameter growth from literature. Original tree ring data were compiled from publicly available ITRDB data,

National Forest Inventories, and data from several tree ring laboratories (see Supplementary Table 3a). In total, we used tree ring records from over 530.000 trees and over 125.000 sites. Maximum ages derived from tree ring records were only included if the field sampling indicated that large trees from natural forests were targeted in the field campaign. Studies were excluded if they focussed purely on even-aged plantations, non-natural systems, non-native species, or if they seemed to consist of partially recorded or artificially truncated time series. Our estimates do not necessarily represent the oldest estimates available, as popular media often reports exaggerated age estimates, which were avoided by relying only on scientific dating methods from traceable sources. In general, tree ring records with less than 10 series were excluded, but we accepted age estimates based on fewer samples if these can reasonably be believed to represent ages close to species' maxima such as published on Oldlists for commonly sampled species (see Supplementary Table 3a). The final database was verified and checked by several authors on this article improving the database. Species names were checked against the World Flora Online (https://www.worldfloraonline.org/downloadData) using the R package WorldFlora, v1.14.5[92]. We only included species listed on the GlobalTreeSearch[93], or those that fit the definition of a tree agreed by IUCN's Global Tree Specialist Groups (GTSG): "a woody plant with usually a single stem growing to a height of at least two metres or if multi-stemmed with then at least one vertical stem five centimetres in diameter at breast height." This database presents a significant extension compared to previous studies (i.e., 438 species[11], 237 species[94], 246 species[34], <200 species[10,27]).

To assess uncertainty on estimates of the maximum longevity for a species, we classified the age estimates for each species into four confidence categories according to the sampling methods, and number of sites and samples included (see Supplementary Table 3c for definitions). Estimates obtained from tree ring records containing less than 150 trees, or series, were classified as the lowest confidence (level 1). Radiocarbon dating on large trees, and growth projections or historical accounts were classified as moderate confidence (level 2). Tree ring records based on more than 150 samples and less than three sites, or those published on Oldlists for commonly sampled species were classified as high confidence (level 3), and tree ring records from more than 3 sites and more than 150 series as very high confidence (level 4). A relatively large portion of species longevity estimates (45%) fell into the lowest confidence category (Supplementary Table 3c). While this could potentially bias our results, we find no evidence for significant changes in the results when this category was excluded from our analysis (see e.g. Supplementary Fig. 3) and we thus chose to present the main results including the maximum number of species. We also note that a relatively large portion of samples (22%) came from growth projections and/or radiocarbon dating, which included important functional groups such as fast-growing tropical pioneer species and extremely old, slow-growing species that generally lack reliable tree ring dating. These age estimates based on projections using short-term growth and mortality rates are subject to several potential biases due to limited data availability, variation in growth and mortality with tree size, and trade-offs between growth and mortality within species that may be poorly captured. However, our results remained largely robust even when excluding growth projections from our analysis although the relationship with wood density for angiosperms disappeared (Supplementary Fig. 4). In all, this provides confidence in our conclusions.

Study sites are well distributed across the globe (see Supplementary Fig. 1a) and included species are from all biomes (Supplementary Fig. 1b). However, the database does not equally represent all biomes and groups. Tree ring data, in particular those from the ITRDB, are overrepresented by old, mostly gymnosperm species growing close to tree lines as these are most valuable for dendro-climate reconstructions[95]. As a result, there is a geographic bias towards

mountain ranges around 40°N latitude such as the American Cordillera, Himalayas, Alps. Tree ring studies further underrepresent tropical trees. To overcome these two biases, we included previously compiled tropical tree ring data[11], and explicitly searched for additional age records for common short-lived species to capture the full spectrum of life history strategies. We note that this resulted in a relatively large set of tropical species growing at temperatures >25 °C and somewhat unbalanced distribution of angiosperm species across the temperature axis (Fig. 1a), but we do not believe this influenced our results. We further note that our dataset sampled only a fraction (534) of the global estimated number of angiosperms species (>70.000)[96]. Filling this gap is a challenge, in particular for tropical angiosperms given the vast number of tropical species and their unreliable formation of tree rings. As a result, significant discussions remain regarding, for example, the longevity of slow-growing tropical shade tolerant species as radiocarbon dating in excess of 1000 years[97] is still regarded uncertain[17,98,99], and obtaining reliable estimates of tropical tree ages remains a major challenge. These shortcomings may raise questions as to what degree the small fraction of sampled angiosperms reflects the spectrum of life history strategies and could have led to biases in tree longevity estimates. To assess whether the life history strategies of the included species is different from the global diversity of strategies, we compared the range of wood densities and maximum tree height for our dataset to that available from global trait databases. This comparison included wood density as it is closely linked to life history variation along the fast-slow axis and other functional attributes of trees[72,74,75] as well as tree height as a proxy for plant stature which is the second most important functional traits axis in trees[37,77,78]. This shows that both the average and range of wood density of our study are very similar to that of the globe for both angiosperm and gymnosperms (Supplementary Fig. 2a, b). It further shows very close agreement with tree height, although the functional trait dataset has a somewhat greater share of short statured trees for angiosperms (Supplementary Fig. 2c, d). This gives us confidence that our sample is a fair reflection of the global life history strategies of trees.

### Environmental data

To assess the effects of climate, environment and soil properties on species' longevity we extracted data from a variety of datasets. Firstly, climate and soil data were obtained, for most species, from the TreeGOER database (https://zenodo.org/records/10008994)[100]. This database combines the Global Biodiversity Information Facility, GBIF, species occurrence data[101] with 30-arc seconds resolution WorldClim 2.1[102], aridity indices such as Climate Moisture Index (CMI)[103], and SoilGrids 2.0 data[104] to provide information on tree species' mean climate and soil environments. For a small number of species (21 out of 739) not included in the TreeGOER database, we used the mean of the WorldClim 2.1 climate data from all sites for that species included in our database. In addition to standard climate and soil metrics, we also extracted growing season length and site level Net Primary Productivity (NPP) from CHELSA V2.1 dataset (https://chelsa-climate.org/downloads/)[105], and calculated species means as the average across all sites included in our database.

For each species, we further calculated species mean proximity to the upper tree line, where tree growth is limited by cold temperatures. This was calculated as the mean of the difference between the actual elevation for each site and the potential upper tree line elevation for that sites estimated using an approximation of the relationship between latitude and potential tree line from Fig. 5 in Paulsen and Korner[106]. Lacking elevation site data were filled in using elevation data from WorldClim 2.1[102].

### Trait data

We extracted species trait data from various databases (see Supplementary Table 4). Maximum tree height was obtained from Tallo[107],

The Gymnosperm Database (www.conifers.org), and from Monumentaltrees.com. Wood density data came from the global wood density database[108], CIRAD wood density[109], complemented with several individual records provided by co-authors or accessed online. Conduit density and size were obtained from refs. 51,63 and 110, and P50 and hydraulic safety margins from the Xylem Functional Traits database[63], complemented with unpublished tropical data (Sunny & Barua, 2024). Leaf traits were obtained from GLOPNet leaf economics dataset[111], and the seedmass data from TRY plant trait database_ request No 30569[112].

To include the maximum number of species and traits combinations in one single analysis, we extracted imputed trait data from ref. 37. These methods predict missing data using Bayesian hierarchical probabilistic matrix factorization algorithms (BHPMF) based on trait-trait correlations and trait variation (see ref. 37 for more information). We used all 17 plant functional traits of ref. 37 and log-transformed the data prior to analysis.

### Analysis

We initially conducted a correlation analysis to evaluate the bivariate relationships between the $\log_{10}$-transformed longevity and various factors, including climate, environment, and traits. We then used multiple regression to assess the relative influence of climate variables, which co-vary at a global scale[37]. We used the Akaike Information Criteria (AIC) to select the set of environmental that best explained variation in longevity for angiosperms and gymnosperms modelled separately. We assessed collinearity between different variables by evaluating pairwise scatter plots between all variables and through the variance inflation factor (VIF). We avoided including closely correlated climate variables (e.g. growing season length and temperature, or temperature and proximity to tree line) within a single model and all VIF were lower than 5. To partition the contributions of different variables to explaining longevity, we used the lmg metric from the R package relaimpo, v2.2.7, as recommended by ref. 113.

We then assessed direct and indirect effects of climate and traits on maximum longevity using Structural Equation Models (SEMs). Previous studies showed that wood density[114], leaf traits[115] and tree height[49,50] all vary along latitudinal gradients of climate and soil. SEM allowed separating direct climate and trait effects on longevity, from indirect effects of climate through climate's effects on traits and radial growth. The included variables in this model were mean annual temperature, precipitation of the driest quarter, and wood density, radial growth ($\log_{10}$-transformed) and maximum tree height. We did not include other trait variables as these were usually available for much fewer species and/or showed poor bivariate relationships (see Supplementary Table 1). We used the R package lavaan, v0.6.19[116] to fit the model, and report on conventional cutoff criteria[117] to evaluate the model fit.

To obtain a more complete overview of trait – longevity relationships, we performed a principal component analysis using the R package FactoMineR, v2.12[118] and 17 commonly used plant trait data from ref. 34. For this analysis all plant trait data were natural log-transformed. All analysis was performed in R 4.4.2[119].

### Evaluation of phylogenetic effects

Longevity is to some degree genetically controlled[1], and evolutionary time between species and their phylogenetic relationships may thus affect our results. We therefore used phylogenetic relationships from ref. 120 to asses phylogenetic signals in both groups and re-analysed the main results accounting for phylogenetic relationships between species using phylogenetic generalised linear models (PGLS) from the R packages caper, v1.0.3[121], phytools, v2.5.2[122] and ape, v5.8.1[123]. We further checked whether climate effects on longevity could be driven by specific taxa selected for particular climates extremes. For example, the *Pinaceae* family contains some of the oldest species, and could be

restricted to cold environments, driving the overall patterns. To check for these effects, we reanalysed the climate longevity relationships using linear mixed effects models from R package lme4[124] incorporating family as a random effect. Finally, it could be argued that well represented genera (e.g. large genus of *Pinus* within the gymnosperms) could dominate the observed patterns. To check for this effect, we reanalysed the results using genus level means of longevity and climate.

## Reporting summary

Further information on research design is available in the Nature Portfolio Reporting Summary linked to this article.

## Data availability

Data on species' maximum longevity, traits, and climate that support the findings of this study are available from https://doi.org/10.6084/m9.figshare.29876984. Original raw tree ring data from the ITRDB can be downloaded from https://www.ncei.noaa.gov/products/paleoclimatology/tree-ring, and tropical tree ring data compilations from https://figshare.com/articles/dataset/Locoselli_et_al_2020_Global_tree-ring_analysis_reveals_rapid_decrease_in_tropical_tree_longevity_with_temperature_PNAS/13119842?file=25178405. Individual longevity records from following oldlists http://www.rmtrr.org/oldlist.htm, https://www.ldeo.columbia.edu/~adk/oldlisteast/, http://www.nativetreesociety.org/dendro/ents_maximum_ages.htm, https://www.oldgrowth.ca/oldtrees/. Tree height data can be downloaded from https://zenodo.org/record/6637599, and maximum height measurements were obtained from https://www.conifers.org and https://Monumentaltrees.com. Wood density data can be obtained from https://zenodo.org/records/13322441, and from https://doi.org/10.18167/DVN1/KRVF0E. Conduit density from https://doi.org/10.5061/dryad.1138, and conduit density, P50 and HSM from https://doi.org/10.5061/dryad.1138, and from https://doi.org/10.1126/sciadv.aav1332. Leaf traits from https://www.nature.com/articles/nature02403#Sec15, and seedmass data from https://www.try-db.org/TryWeb/dp.php, database request No 30569. Mean climate and soil data for a species were obtained from the TreeGOER database https://zenodo.org/records/10008994, and gridded climate and elevation data from https://www.worldclim.org/data/worldclim21.html, growing season length and site level Net Primary Productivity (NPP) from https://chelsa-climate.org/. Species occurrence data from https://doi.org/10.15468/dl.77gcvq.

## Code availability

Code to reproduce the Figs. 1–3 and Supplementary Figs. 3–6, 8, 9 and statistics are available from https://doi.org/10.6084/m9.figshare.29876984.

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

## Acknowledgements

We acknowledge the contributors to International Tree-Ring Data Bank for making available raw tree-ring data, and we thank staff at the Direction des Inventaires Forestiers of the Ministère des Ressources naturelles et des Forêts du Québec for sharing tree-ring and sample plot data from the forest inventory program in Quebec, Canada. We further thank Ailene Ettinger, Gregory Peterson, Janneke Hille Ris Lambers, Jeremy Little, Jill Harvey, and Jordi Axelsons for contributing original data. This study was supported by the following grants; National Environmental Research Council grants NE/S008659/1 (R.B.), NE/N012542/1 (E.G.), and NE/R005079/1 (E.G., R.S.); FAPESP grants 12/50457-4, 2019/08783-0 (G.L., G.C.) and 17/5008-3 (G.L., G.C.); Conselho Nacional de Desenvolvimento Científico e Tecnológico, CNPq, grants 478503/2009 (G.L., G.C.), 311247/2021-0 (J.S.) and 441811/2020-5 (J.S.); CNPq/

FAPEAM, Fundação de Amparo à Pesquisa do Estado do Amazonas, grant number 01.02.016301.02630/2022-76 (J.S.); Czech Science Foundation research grants 24-12210 K (J.P. and M.S.) and 23-05272S (J.A., J.D., K.K., N.A., P.F., V.B.); Mobility Plus between the Czech Republic and Taiwan, NSTC-24-08 (J.A., J.D., K.K., N.A., P.F., V.B.); Czech Academy of Sciences long-term research development project No. RVO 67985939 (J.A., J.D., K.K., N.A., P.F., V.B.); Utah Agricultural Experiment Station, Utah State University, and approved as journal paper number 9803 (R.J.D.); Academy of Finland, #339788 (S.H.); European Union, NextGenerationEU, Italian Ministry of University and Research under PNRR - M4C2-I1.4 Project code: CN00000033, Title: NBFC - National Biodiversity Future Center, CUP: J83C22000860007 (G.P.); Ministry of University and Research (MUR) via the Agritech National Research Centre, European Union Next-GenerationEU PNRR M4C2-I1.4 Project Code: CN00000022 (A.D.); Departments of Excellence (Law 232/2016) Project 2023-27 "Digital, Intelligent, Green and Sustainable (D.I.Ver.So)" (A.D.); National Science Foundation, Division of Environmental Biology, award #1945910 (N.P.); Directorate for Biological Sciences, Emerging Frontiers, award #1241870 (N.P.); Redes Federales de Alto Impacto, Bosque-Clima CN32 (L.L., R.V.); MSMT INTER-EXCELLENCE, # LUAUS24258 (J.D.), Estonian Research Council, grant PSG1044 (J.A.).

## Author contributions

R.B., G.L., S.K., E.G., and R.W. designed the study, R.B., R.W. and S.K. downloaded and compiled functional traits and ITRDB datasets, R.B., R.W. and S.K. analysed data, G.L. and S.K. compiled the tropical longevity datasets, M.M., D.B., R.S. and P.R. provided functional traits data, R.B., G.L., S.K., S.V., C.E., G.P. and N.P. revised and improved the longevity database, R.B., G.L., S.V., J.A., N.A., L.A., M.B., V.B., B.B., P.B., G.C., J.dR., J.V.D., A.D., J.D., L.D., C.E., P.F., H.G., S.H., S.Kl., K.K., D.L., S.L., L.L., T.N., J.P., N.P., G.P., C.R., D.S., J.S., J.D.S., D.S., M.S., R.V., L.W., and C.Z. contributed original longevity data, R.B. wrote the first draft of the manuscript and all authors revised the manuscript.

## Competing interests

The authors declare no competing interests.

## Additional information

Roel J. W. Brienen [1] ✉, Giuliano Maselli Locosselli[2], Stefan Krottenthaler[3], Emanuel Gloor[1], Robyn Wrigley[4], Steven L. Voelker[5], Jan Altman [6,7,8], Nela Altmanova[6,9], Leander D. L. Anderegg [10], Michele Baliva[11], Deepak Barua [12], Vaclav Bazant[7], Bryan Black[13], Peter M. Brown[14], Gregorio Ceccantini[15], R. Justin DeRose[16], Jose Villanueva Diaz[17], Alfredo Di Filippo [18], Jiri Dolezal [6,9], Louis Duchesne[19], Christopher Earle [20], Pavel Fibich [6,9], Hardy Griesbauer[21], Samuli Helama [22], Stefan Klesse [23], Kirill Korznikov[6], David Lindenmayer [24], Shuhui Liu[25], Lidio Lopez [26], Maurizio Mencuccini [27,28], Thomas A. Nagel [29], Jakob Pavlin [7], Neil Pederson [30,31], Gianluca Piovesan [11], Christina Restaino[32], Peter B. Reich [33,34], David Sauchyn [35], Jochen Schöngart[36], John D. Shaw [37], Dan Smith[38], Ron Sunny[39], Miroslav Svoboda [7], Ricardo Villalba [26], Lisa J. Wood[40] & Chunyu Zhang [25]

[1]School of Geography, University of Leeds, Leeds, UK. [2]Center of Nuclear Energy in Agriculture, University of São Paulo, Piracicaba, Brazil. [3]Physical Geography, University of Passau, Passau, Germany. [4]School of Earth and Environment, University of Leeds, Leeds, UK. [5]College of Forest Resources and Environmental Science, Michigan Technological University, Houghton, MI, USA. [6]Institute of Botany of the Czech Academy of Sciences, Třeboň, Czech Republic. [7]Faculty of Forestry and Wood Sciences, Czech University of Life Sciences, Prague, Czech Republic. [8]Department of Geography, Institute of Ecology and Earth Sciences, University of Tartu, Tartu, Estonia. [9]Faculty of Science, University of South Bohemia, České Budějovice, Czech Republic. [10]Department of Ecology, Evolution & Marine Biology, University of California Santa Barbara, Santa Barbara, CA, USA. [11]Department of ecological and biological science (DEB), Università della Tuscia, Viterbo, Italy. [12]Department of Biology, Indian Institute of Science Education and Research, Pune, India. [13]Laboratory of Tree Ring Research, University of Arizona, Tucson, AZ, USA. [14]Rocky Mountain Tree-Ring Research, Fort Collins, CO, USA. [15]Department of Botany, University of São Paulo, Institute of Biosciences, São Paulo, SP, Brazil. [16]Department of Wildland Resources and Ecology Center, Logan, UT, USA. [17]Laboratorio de Dendrocronologia, Instituto Nacional de Investigaciones Forestales, Agricolas y Pecuarias, Gomez Palacio, Mexico. [18]Department of Agriculture and Forest Science (DAFNE), Università della Tuscia, Viterbo, Italy. [19]Ministère des Ressources naturelles et des Forêts, Direction de la recherche forestière, Quebec city, QC, Canada. [20]Gymnosperm Database, Olympia, WA, USA. [21]British Columbia Ministry of Forests, Prince George, BC, Canada. [22]Natural Resources Institute Finland, Rovaniemi, Finland. [23]Forest and Soil Ecology, Swiss Federal Institute for Forest, Snow and Landscape Research WSL, Birmensdorf, Switzerland. [24]Fenner School of Environment and Society, The Australian National University, Canberra, ACT, Australia. [25]Research Center of Forest Management

Engineering of State Forestry and Grassland Administration, Beijing Forestry University, Beijing, China. [26]Laboratorio de Dendrocronología e Historia Ambiental IANIGLA/CONICET, Mendoza, Argentina. [27]CREAF, Bellaterra, Spain. [28]ICREA, Barcelona, Spain. [29]Department of forestry and renewable forest resources, University of Ljubljana, Ljubljana, Slovenia. [30]Independent Scholar, Maynard, MA, USA. [31]Harvard Forest, Harvard University, Petersham, MA, USA. [32]University of Nevada, Reno, Reno, NV, USA. [33]Institute for Global Change Biology, University of Michigan, Ann Arbor, MI, USA. [34]Department of Forest Resources, University of Minnesota, St. Paul, MN, USA. [35]Prairie Adaptation Research Collaborative, Geography and Environmental Studies, University of Regina, Regina, Canada. [36]Instituto Nacional de Pesquisas da Amazônia (INPA), Ecologia, Monitoramento e Uso Sustentável de Áreas Úmidas (MAUA), Manaus, AM, Brazil. [37]Rocky Mountain Research Station, USDA Forest Service, Ogden, UT, USA. [38]Department of Geography, University of Victoria, Victoria, BC, Canada. [39]Department of Botany, St Joseph's College (Autonomous), Devagiri, Calicut, Kerala, India. [40]University of Northern British Columbia, Faculty of Environment, Prince George, BC, Canada. ✉e-mail: r.brienen@leeds.ac.uk

