## [Transparent Peer Review file · Nature Communications]

Contrasting pathways to tree longevity in gymnosperms and angiosperms

Corresponding Author: Dr Roel Brien

A version of this paper was originally rejected for publication by Nature Communications, however that decision was reconsidered after appeal by the authors.

Version 0:

Reviewer comments:

Reviewer #1

(Remarks to the Author)

The authors of this piece on tree longevity did a great job in revising the original text. The response letter impressed me and removes most of my concerns. The text was adjusted accordingly and I support publication.

The rest is opinion, and I respect the authors to retain theirs. What remains, is my uneasiness about placing those gymnosperm taxa well known for their great tree age and belonging to warm, humid climates into an exception category. In my view the cold = old conclusion results from attributing the same (statistical) weight to often single species ancient genera as to species in multispecies genera such as *Pinus* and *Picea*. At genus level, the cold = old hypothesis would most likely become falsified. So, I simply do not believe that cold climates select for longevity. These climates select for instance for pines, and some of these can get very old as well, but statistically, they dominate the analysis because of their many species (a within-gymnosperms phylogenetic bias). I did not see this issue so clearly in the first round, as I see it now. Instead of 'cold' (line 507), I rather prefer this statement copied from the revised text (line 194-195):

"We further find that gymnosperms show a decrease in longevity with increasing site productivity, whereas angiosperms show a weak positive relationship with potential site productivity (Fig, 2d)."

...leaving it open whether temperature, moisture, nutrients, soil chemistry are decisive. In Junipers it might be drought tolerance. In essence, it is a high rate of radial growth that selects against long life. Very important.

Another important statement is this one (line 274-275):

"...the existence of relatively weak effects of functional traits on tree longevity, once phylogenetic differences between angiosperms and gymnosperms are taken into account".

I hope many people will get this message.

I suggest to remove the word rainforest in line 447, it does not fit *Sequoiadendron*, nor *Agathis*, nor *Fitzroya*, and I doubt whether coastal redwoods would be ranked as rainforest taxa by experts. To my knowledge, the only rain forest in the US Westcoast is on the Olympia peninsula, where *Thuja*-giants dominate. *Cryptomeria* is certainly belonging to warm humid temperate. So, 'Humid' might do...

So, I am fine with the revision, the phylogenetic bias story (except for the above aspect) is treated adequately and so are trait syndromes and the novelty question that referee 2 rose. Was glad to see the omissions that removed doubtful aspects. It is not relevant here that I disbelieve in hydraulics exerting a decisive role (I think the 'exceptions' falsify that story, e.g. E. regnans, the 80 m E-Asia Dipterocarps), but this is a valid element of the discussion and not a result of the analysis.

Offering the revision in tracking mode helped a lot in re-visiting this text.
Overall this is a most valid analysis, and I am sure it will stir up discussion.
No need to consult with me again.
Christian Körner

Reviewer #2

(Remarks to the Author)

This manuscript was transferred from NEE. The authors have responded all the concerns from both reviewers including me. However, their responses cannot convince me for the "novel findings" of this study, and my concerns still remain.

1. Novelty. The authors summarized their novelty in their response to my review comment as "We identify two different ways for trees to become old. In addition to the previously identified "adversity pathway", we find that species in humid, productive environments exhibit greater longevity (a "productivity pathway"). And we find that these pathways differ for angiosperms and gymnosperms". Here the authors admit that the "adversity path way" was "previously identified". Their new finding is "productivity pathway" (precipitation-productivity-longevity). However, the cited literature (reference 11, Locosselli, G. M. et al., 2020, PNAS) clearly stated that "we find consistent decreases in tree longevity with increasing aridity". I therefore conclude that the novelty of the manuscript is very limited. The authors argued that they found angiosperms and gymnosperms follow different pathways. However, they set out a subgroup that are mainly temperate conifers with long lifespan. Therefore, they cannot fully distinguish between angiosperms and gymnosperms.

2. Phylogeny and evolution. The authors admitted that "longevity is, at least to some degree, inherent to a plant species" in their responses. However, they stressed that "... is poorly understood but is beyond the scope of the present work" in the manuscript. The aim of this manuscript is to detect "climate-longevity relationships" "across species". However, this may cause huge controversy, as also reminded by another reviewer for many times with many examples. They respond that "the species' growth rates and longevity are all the outcome of environmental and climatic filtering of species and long evolutionary selection process that has resulted in an optimisation of their specific traits (including demographic traits such as growth and longevity) for that particular environment", therefore, the authors have to take into consideration of the different "long evolutionary selection process" across species.

In their response, they stated ".....an optimisation of their specific traits (including demographic traits such as growth and longevity) for that particular environment", but they did not admit that "The fact that we do not find strong relationships between longevity and traits within groups does not mean that our approach has failed". Such kinds of contradictions can also be found in the manuscript, which makes the manuscript not logically organized.

3. Climate change. The authors responded that "We realised this section was outside the scope of the article, and it was also criticised by Reviewer 1. We have now deleted this entire section". But in the introduction, the following sentence remains: "These insights can support predictions of future responses to global change, for example, by providing Earth System Models with more realistic estimates of tree turnover" (Line 96-98). However, the climate and longevity relationships are mainly correlations, as shown in all the three figures in the main text. Do you really believe that your correlations between climate and longevity can contribute to Earth System Models with more realistic estimates of tree turnover?

4. Conclusion. The authors responded that I had misunderstood their conclusion and emphasized in their responses that "when pooling the angiosperms and gymnosperms most traits were significantly related to longevity (first left columns), but these relationships disappeared, or became weaker, when the analysis was done per group (columns to the right). We go from 12 significant trait longevity relationships (when the two groups are pooled) to much fewer relationships for the groups separately (i.e., 4 for Angiosperms and 2 for conifers)". Do you mean "pooling the angiosperms and gymnosperms" is better than separating them in explaining the role of traits? If so, why did you separate them?

Version 2:

Reviewer comments:

Reviewer #3

(Remarks to the Author)

This manuscript addresses important questions related to variation in tree longevity and its ecological drivers. I recognize that this work has already undergone prior rounds of peer review, which have helped resolve many technical details and improve clarity. The authors' responses demonstrate significant effort to strengthen the manuscript, and I agree that the study presents novel insights, particularly the identification of two contrasting pathways to tree longevity across environmental gradients.

Nevertheless, I believe some conceptual uncertainties remain, which are inherently difficult to resolve fully but are worth considering. In addition, I would like to raise another of these difficult-to-address issues, which could have implications for

the interpretation of the results. Given that gymnosperms tend to be more prevalent in environments subjected to freezing temperatures, I wonder whether similar relationships between temperature, precipitation, and longevity would emerge if gymnosperms and angiosperms were pooled and analyzed collectively, but separately for environments with and without freezing conditions. Such an approach could help disentangle whether the observed patterns are primarily driven by phylogenetic differences, environmental filtering, or their interaction.

In summary, I believe the manuscript, as it stands, contributes valuable and novel perspectives to ongoing research on variation in tree longevity.

Reviewer #4

(Remarks to the Author)

I provide comments in the attached document.

Version 3:

Reviewer comments:

Reviewer #3

(Remarks to the Author)

I greatly appreciate the additional analysis separating species into freezing versus frost-free environments, along with its thoughtful integration into the discussion. Importantly, the authors acknowledge that uncertainties remain regarding the extent to which observed differences are driven by environmental filtering versus intrinsic phylogenetic effects, which adds valuable nuance and depth to the manuscript.

I am fully satisfied with the authors' revisions and responses. I congratulate them on this significant contribution and look forward to the publication of the article, which will be an important resource to cite.

Reviewer #4

(Remarks to the Author)

This is a strong paper, and the final round of revisions was generally thorough and appropriate. I only have one final minor comment. The authors stated that their results were robust to the exclusion of the growth-based and mortality-based projections of tree longevity, but omitting these data actually caused the association between wood density and Angiosperm longevity to become non-significant (Supplementary Fig. 4e). This should be stated explicitly in the results (lines 200-201) so that it is clear for the reader that this association does not hold up when those data are excluded.

Point-by-point response to reviewers.

Reviewer #1 (Remarks to the Author):

The authors of this piece on tree longevity did a great job in revising the original text. The response letter impressed me and removes most of my concerns. The text was adjusted accordingly and I support publication.

The rest is opinion, and I respect the authors to retain theirs. What remains, is my uneasiness about placing those gymnosperm taxa well known for their great tree age and belonging to warm, humid climates into an exception category. In my view the cold = old conclusion results from attributing the same (statistical) weight to often single species ancient genera as to species in multispecies genera such as *Pinus* and *Picea*. At genus level, the cold = old hypothesis would most likely become falsified. So, I simply do not believe that cold climates select for longevity. These climates select for instance for pines, and some of these can get very old as well, but statistically, they dominate the analysis because of their many species (a within-gymnosperms phylogenetic bias). I did not see this issue so clearly in the first round, as I see it now. Instead of 'cold' (line 507), I rather prefer this statement copied from the revised text (line 194-195):

“We further find that gymnosperms show a decrease in longevity with increasing site productivity, whereas angiosperms show a weak positive relationship with potential site productivity (Fig, 2d).”

...leaving it open whether temperature, moisture, nutrients, soil chemistry are decisive. In Junipers it might be drought tolerance. In essence, it is a high rate of radial growth that selects against long life. Very important.

>> We understand this concern, and we have checked how observed relationships hold up at genus level by plotting “genus level” mean longevity against the two key climate variables (temperature and precipitation of the driest quarter, see Supplementary Figure 7).

Supplementary Figure 7. Relationships between longevity and the two most important climate variables at genus level. For each genus we calculated the mean longevity and mean temperature (a) and precipitation during the driest quarter (b). These results show that the key patterns are not driven simply by few widespread genera showing strong relationships with climate but hold up at genus level.

Another important statement is this one (line 274-275):

“...the existence of relatively weak effects of functional traits on tree longevity, once phylogenetic differences between angiosperms and gymnosperms are taken into account”.

I hope many people will get this message.

>> We think readers will get this message as it is formulated clearly at the start of the discussion, and again at the end (i.e. we say, “*In summary, our analyses showed that most species-specific traits provided relatively weak relationships with species’ tree longevity within both groups.*”).

I suggest to remove the word rainforest in line 447, it does not fit *Sequoiadendron*, nor *Agathis*, nor *Fitzroya*, and I doubt whether coastal redwoods would be ranked as rainforest taxa by experts. To my knowledge, the only rain forest in the US Westcoast is on the Olympia peninsula, where *Thuja*-giants dominate. *Cryptomeria* is certainly belonging to warm humid temperate. So, ‘Humid’ might do...

>> We agree and have changed this to “humid forests”.

So, I am fine with the revision, the phylogenetic bias story (except for the above aspect) is treated adequately and so are trait syndromes and the novelty question that referee 2 rose. Was glad to see the omissions that removed doubtful aspects. It is not relevant here that I disbelieve in hydraulics exerting a decisive role (I think the ‘exceptions’ falsify

that story, e.g. E. regnans, the 80 m E-Asia Dipterocarps), but this is a valid element of the discussion and not a result of the analysis.

>> Please see our additional work where we account for phylogeny.

Offering the revision in tracking mode helped a lot in re-visiting this text.
Overall this is a most valid analysis, and I am sure it will stir up discussion.
No need to consult with me again.
Christian Körner

Reviewer #2 (Remarks to the Author):

This manuscript was transferred from NEE. The authors have responded all the concerns from both reviewers including me. However, their responses cannot convince me for the “novel findings” of this study, and my concerns still remain.

1. Novelty. The authors summarized their novelty in their response to my review comment as “We identify two different ways for trees to become old. In addition to the previously identified “adversity pathway”, we find that species in humid, productive environments exhibit greater longevity (a “productivity pathway”). And we find that these pathways differ for angiosperms and gymnosperms”. Here the authors admit that the “adversity path way” was “previously identified”. Their new finding is “productivity pathway” (precipitation-productivity-longevity). However, the cited literature (reference 11, Locosselli, G. M. et al., 2020, PNAS) clearly stated that “we find consistent decreases in tree longevity with increasing aridity”. I therefore conclude that the novelty of the manuscript is very limited. The authors argued that they found angiosperms and gymnosperms follow different pathways. However, they set out a subgroup that are mainly temperate conifers with long lifespan. Therefore, they cannot fully distinguish between angiosperms and gymnosperms.

>> We strongly contest this assertion. Both the distinction of the two pathways for the two groups and the identification of the mechanism using statistical attribution are novel. The mentioned study by Locosselli et al. (2020) did not present separate analysis for angiosperms and gymnosperms, and this is thus a new finding. The description of a subgroup of temperate conifers deviating from this pattern does not invalidate the general differences between groups as the main analysis of the gymnosperms presented in the manuscript included this subgroup. Thus the patterns are robust. Another novelty that we mentioned in our response is the identification of mechanisms driving tree longevity such as slow growth in conifers under low temperature, and increased tree height and competition under humid conditions for angiosperms. While we cite some of the first studies proposing the adversity mechanism, no study before has shown this for a large dataset like the one presented here, and basically no study before has shown the mechanism(s) for the second pathway (mainly found in Angiosperms).

2. Phylogeny and evolution. The authors admitted that “longevity is, at least to some degree, inherent to a plant species” in their responses. However, they stressed that “... is poorly understood but is beyond the scope of the present work” in the manuscript. The aim of this manuscript is to detect “climate-longevity relationships” “across species”. However, this may cause huge controversy, as also reminded by another

reviewer for many times with many examples. They respond that “the species’ growth rates and longevity are all the outcome of environmental and climatic filtering of species and long evolutionary selection process that has resulted in an optimisation of their specific traits (including demographic traits such as growth and longevity) for that particular environment”, therefore, the authors have to take into consideration of the different “long evolutionary selection process” across species.

In their response, they stated “.....an optimisation of their specific traits (including demographic traits such as growth and longevity) for that particular environment”, but they did not admit that “The fact that we do not find strong relationships between longevity and traits within groups does not mean that our approach has failed”. Such kinds of contradictions can also be found in the manuscript, which makes the manuscript not logically organized.

>> We are not quite sure what the contradiction is, but we have now reanalysed the key results taking fully into account phylogeny between species. While it shows that there is a phylogenetic signal in tree longevity, the strength of the relationships remains the same as the analysis presented in our original manuscript (see Supplementary Table 5). Our conclusions and findings thus remain robust. See also our arguments in the appeal letter.

3. Climate change. The authors responded that “We realised this section was outside the scope of the article, and it was also criticised by Reviewer 1. We have now deleted this entire section”. But in the introduction, the following sentence remains: “These insights can support predictions of future responses to global change, for example, by providing Earth System Models with more realistic estimates of tree turnover” (Line 96-98). However, the climate and longevity relationships are mainly correlations, as shown in all the three figures in the main text. Do you really believe that your correlations between climate and longevity can contribute to Earth System Models with more realistic estimates of tree turnover?

>> This seems a minor comments, but we do believe that empirical estimates of tree turnover rates and their variation across biomes (as we show in our manuscript, see Fig. 1b), can help Earth System models. This is what the modelling community themselves clearly state. For example, the study by Friend et al. (2014) that we cite, states that “*Further work should focus on confronting the processes and emergent model behavior responsible for these discontinuities with observational data. Spatial and temporal variability in carbon residence time and tree mortality is an obvious place to start, and although recent studies have identified important sources of relevant data (e.g., refs. 20 and 21), data on key processes such as mortality at large scales are rare.*”

We believe it is not such a stretch to state this.

4. Conclusion. The authors responded that I had misunderstood their conclusion and emphasized in their responses that “when pooling the angiosperms and gymnosperms most traits were significantly related to longevity (first left columns), but these relationships disappeared, or became weaker, when the analysis was done per group (columns to the right). We go from 12 significant trait longevity relationships (when the two groups are pooled) to much fewer relationships for the groups separately (i.e., 4 for

Angiosperms and 2 for conifers)”. Do you mean "pooling the angiosperms and gymnosperms" is better than separating them in explaining the role of traits? If so, why did you separate them?

>> We mean that when pooling all data of angiosperms and gymnosperms we do find that all traits (except for one) show significant correlations with longevity. One could indeed thus conclude that traits explain more longevity variation when data are pooled. Why did we separate them? We separated them firstly because we know from literature that the two groups have very different traits, growth strategies and longevities, and this is thus not a new finding. Secondly, we wanted to avoid taxonomic biases. We make this clear at the end of the introduction where we say “*Previous studies showed that angiosperms and gymnosperms differ strongly in their growth strategies²⁹, photosynthetic capacity³⁰, hydraulic traits³¹⁻³³ and longevity^{1,10,11}, which we confirm in our analysis of the full dataset. Given that these taxonomic groups diverged over 300 million years ago³⁴ and that they differ in their global distribution³⁵, we primarily analyse the results for these groups separately to avoid any phylogenetic bias in the results.*“

Additional text to manuscript:

Full text below and table and Supplementary figure 5 are added to the manuscript.

Evaluation of phylogenetic effects

Longevity is to some degree genetically controlled¹, and evolutionary time between species and their phylogenetic relationships may thus affect our results. We used phylogenetic relationships from ref.118 to reveal significant phylogenetic signals in both groups (Pagel's λ for Gymnosperms = 0.905, $p < 0.001$; Pagel's λ for Angiosperms: 0.46, $p < 0.001$). We then performed additional checks to ensure our results are robust with regard to these phylogenetic relationships. To this end, we first re-analysed the main results accounting for phylogenetic relationships between species using phylogenetic generalised linear models (PGLS) from the R packages *caper*¹¹⁹, *phytools2.0*¹²⁰ and *ape 5.0*¹²¹. This shows that core relationships with climate, growth, and maximum tree height all remain significant with very little change in the correlation strength (see Supplementary table 5). The only correlation that switched to marginally significant ($p = 0.069$) is that between angiosperm longevity and wood density. Secondly, we checked whether climate effects on longevity could be driven by specific taxa selected for particular climates extremes. For example, the *Pinaceae* family contains some of the oldest species, and could be restricted to cold environments, driving the overall patterns. Reanalysis of the data using linear mixed effects models from R package *lme4*¹²² incorporating family as a random effect proves that relationships with climate (i.e., temperature for Gymnosperms and precipitation for angiosperms) hold up within families. Finally, it could be argued that well represented genera (e.g. large genus of *Pinus* within the gymnosperms) could dominate the observed patterns. To check for this effect, we reanalysed the results using genus level means of longevity and climate. It confirms again that longevity – climate relationships remain robust for both groups (Supplementary Fig. 7a,b).

Supplementary Table 5. Comparison of standard and phylogenetically controlled correlations for the key variables controlling variation in longevity (cf. Supplementary Table 1). Standard correlations are Pearson's correlations, and phylogenetic controlled correlation were performed using phylogenetic generalised linear models (PGLS) (see methods).

	Angiosperms						Gymnosperms					
	Standard correlation			Phylogenetically controlled			Standard correlation			Phylogenetically controlled		
	R	p value	n	R	p value	n	R	p value	n	R	p value	n
Mean annual temperature	0.030	0.488	534	0.038	0.405	476	-0.378	0.000	205	-0.420	0.000	200
Precipitation driest quarter**	0.332	0.000	488	0.277	0.000	432	0.080	0.258	203	0.072	0.312	198
Mean radial growth*	-0.564	0.000	429	-0.507	0.000	381	-0.567	0.000	184	-0.467	0.000	179
Maximum tree height	0.351	0.000	325	0.294	0.000	291	0.200	0.005	198	0.261	0.000	193
Wood density	0.124	0.009	441	0.091	0.069	399	-0.081	0.333	144	0.123	0.146	140

* These data were \log_{10} -transformed for the analysis; ** Correlations with precipitation and Climate moisture index exclude species that have their main occurrence in floodplains or wetlands.

Point-by-point response to reviewers

Reviewer #1 (Remarks to the Author):

The authors of this piece on tree longevity did a great job in revising the original text. The response letter impressed me and removes most of my concerns. The text was adjusted accordingly and I support publication.

The rest is opinion, and I respect the authors to retain theirs. What remains, is my uneasiness about placing those gymnosperm taxa well known for their great tree age and belonging to warm, humid climates into an exception category. In my view the cold = old conclusion results from attributing the same (statistical) weight to often single species ancient genera as to species in multispecies genera such as *Pinus* and *Picea*. At genus level, the cold = old hypothesis would most likely become falsified. So, I simply do not believe that cold climates select for longevity. These climates select for instance for pines, and some of these can get very old as well, but statistically, they dominate the analysis because of their many species (a within-gymnosperms phylogenetic bias). I did not see this issue so clearly in the first round, as I see it now. Instead of 'cold' (line 507), I rather prefer this statement copied from the revised text (line 194-195):

"We further find that gymnosperms show a decrease in longevity with increasing site productivity, whereas angiosperms show a weak positive relationship with potential site productivity (Fig, 2d)."

...leaving it open whether temperature, moisture, nutrients, soil chemistry are decisive. In Junipers it might be drought tolerance. In essence, it is a high rate of radial growth that selects against long life. Very important.

>> We strongly agree that high growth selects against long life. To leave it more open whether it is soil or climate, we now changed line 507 to "*sites with adverse growing conditions (i.e., low temperature, short growing seasons, poor soils, etc.)*."

As for potential bias due to disproportionate weight to specific taxa, we have now also checked how observed relationships hold up at genus level by plotting "genus level" mean longevity against the two key climate variables (temperature and precipitation of the driest quarter). This shows that the relationships are holding up at genus level. We have added this figure to the supplementary info and also added following text to the results section:

"Finally, we reanalysed the results using genus level means of longevity and climate. It confirms again that longevity-climate relationships remain robust within both groups (Supplementary Fig. 7a,b)."

Supplementary Figure 7. Relationships between longevity and the two most important climate, variables at genus level. For each genus we calculated the mean longevity and mean temperature (a) and precipitation during the driest quarter (b). These results show that the key patterns are not driven simply by few widespread genera showing strong relationships with climate but hold up at genus level.

Another important statement is this one (line 274-275):

“...the existence of relatively weak effects of functional traits on tree longevity, once phylogenetic differences between angiosperms and gymnosperms are taken into account”.

I hope many people will get this message.

>> We think readers will get this message as it is formulated clearly at the start of the discussion, and again at the end (ie we say , “*In summary, our analyses showed that most species-specific traits provided relatively weak relationships with species’ tree longevity within both groups.*”).

I suggest to remove the word rainforest in line 447, it does not fit Sequoiadendron, nor Agathis, nor Fitzroya, and I doubt whether coastal redwoods would be ranked as rainforest taxa by experts. To my knowledge, the only rain forest in the US Westcoast is on the Olympia peninsula, where Thuja-giants dominate. Cryptomeria is certainly belonging to warm humid temperate. So, ‘Humid’ might do...

>> We agree and have changed this to “humid forests”.

So, I am fine with the revision, the phylogenetic bias story (except for the above aspect) is treated adequately and so are trait syndromes and the novelty question that referee 2 rose. Was glad to see the omissions that removed doubtful aspects. It is not relevant here that I disbelieve in hydraulics exerting a decisive role (I think the ‘exceptions’ falsify that story, e.g. E. regnans, the 80 m E-Asia Dipterocarps), but this is a valid element of the discussion and not a result of the analysis.

>> We added a full analysis on the effects of phylogeny to the manuscript. See our response to comment#2 below of reviewer 2.

Offering the revision in tracking mode helped a lot in re-visiting this text. Overall this is a most valid analysis, and I am sure it will stir up discussion. No need to consult with me again.
Christian Körner

Reviewer #2 (Remarks to the Author):

This manuscript was transferred from NEE. The authors have responded all the concerns from both reviewers including me. However, their responses cannot convince me for the “novel findings” of this study, and my concerns still remain. 1. Novelty. The authors summarized their novelty in their response to my review comment as “We identify two different ways for trees to become old. In addition to the previously identified “adversity pathway”, we find that species in humid, productive environments exhibit greater longevity (a “productivity pathway”). And we find that these pathways differ for angiosperms and gymnosperms”. Here the authors admit that the “adversity path way” was “previously identified”. Their new finding is “productivity pathway” (precipitation-productivity-longevity). However, the cited literature (reference 11, Locosselli, G. M. et al., 2020, PNAS) clearly stated that “we find consistent decreases in tree longevity with increasing aridity”. I therefore conclude that the novelty of the manuscript is very limited. The authors argued that they found angiosperms and gymnosperms follow different pathways. However, they set out a subgroup that are mainly temperate conifers with long lifespan. Therefore, they cannot fully distinguish between angiosperms and gymnosperms.

>> We contest this assertion. Both the distinction of the two pathways for the two groups and the identification of the mechanism using statistical attribution are novel. The mentioned study by Locosselli et al. (2020) did not present separate analysis for angiosperms and gymnosperms, and this is thus a new finding. The description of a subgroup of temperate conifers deviating from this pattern does not invalidate the general differences between groups as the main analysis of the gymnosperms presented in the manuscript included this subgroup and thus the overall patterns are robust.

Another novelty that we mentioned in our response is the identification of mechanisms driving tree longevity such as slow growth in conifers under low temperature, and increased tree height and competition under humid conditions for angiosperms.

While we cite some of the first studies proposing the adversity mechanism, no study before has shown this for a large dataset like the one presented here, and no study before has shown the mechanism(s) for the second pathway (mainly found in Angiosperms) in as we do here using robust statistical approaches (eg. structural equation modelling, SEM, see our Figure 3).

2. Phylogeny and evolution. The authors admitted that “longevity is, at least to some degree, inherent to a plant species” in their responses. However, they stressed that “... is poorly understood but is beyond the scope of the present work” in the manuscript. The aim of this manuscript is to detect “climate-longevity relationships” “across species”. However, this may cause huge controversy, as also reminded by another reviewer for many times with many examples. They respond that “the species’ growth rates and longevity are all the outcome of environmental and climatic filtering of species and long evolutionary selection process that has resulted in an

optimisation of their specific traits (including demographic traits such as growth and longevity) for that particular environment”, therefore, the authors have to take into consideration of the different “long evolutionary selection process” across species. In their response, they stated “.....an optimisation of their specific traits (including demographic traits such as growth and longevity) for that particular environment”, but they did not admit that “The fact that we do not find strong relationships between longevity and traits within groups does not mean that our approach has failed”. Such kinds of contradictions can also be found in the manuscript, which makes the manuscript not logically organized.

>> We are not quite sure what the contradiction is, but agree on the importance of phylogeny and now analysed to what degree that phylogeny could have affected our results. This analysis shows a phylogenetic signal in tree longevity -as the reviewer suggested- and we thus reanalysed the key results taking into account phylogeny between species. This shows that the strength of the relationships remains largely the same as the analysis presented in our original manuscript, and shows that our conclusions and findings remain robust.

We have added these new analyses to the methods and results sections of the manuscript and copy the text of these sections below.

METHODS:

*Longevity is to some degree genetically controlled¹, and evolutionary time between species and their phylogenetic relationships may thus affect our results. We therefore used phylogenetic relationships from ref.² to assess phylogenetic signals in both groups and re-analysed the main results accounting for phylogenetic relationships between species using phylogenetic generalised linear models (PGLS) from the R packages *caper*³, *phytools*2.0⁴ and *ape* 5.0⁵. We further checked whether climate effects on longevity could be driven by specific taxa selected for particular climates extremes. For example, the Pinaceae family contains some of the oldest species, and could be restricted to cold environments, driving the overall patterns. To check for these effects, we reanalysed the climate longevity relationships using linear mixed effects models from R package *lme4*⁶ incorporating family as a random effect. Finally, it could be argued that well represented genera (e.g. large genus of *Pinus* within the gymnosperms) could dominate the observed patterns. To check for this effect, we reanalysed the results using genus level means of longevity and climate.*

RESULTS :

We find that both groups show significant phylogenetic signals in longevity variation (Pagel's λ for Gymnosperms = 0.905, $p < 0.001$; Pagel's λ for Angiosperms: 0.46, $p < 0.001$). We therefore checked whether our results are robust with regard to these phylogenetic relationships. To this end, we first repeated the correlation analysis for key variables shown in Figure 2 whilst accounting for phylogenetic relationships between species and find that all remain significant with very little change in the correlation strength (see Table 2). The only correlations that switched to non-significant are those between angiosperm longevity and wood density ($p = 0.069$) and potential site productivity ($p = 0.108$). Secondly, we reanalysed the data using linear mixed effects models incorporating family as a random effect. This proves that relationships with climate hold up within families (i.e., standardized effect of temperature on gymnosperm longevity = -0.157, t -value = -7.23, $p < 0.001$, $n = 205$, standardized effect of precipitation during the driest quarter on angiosperm longevity = +0.107, t -value = 6.43, $p < 0.001$, $n = 534$). Finally, we reanalysed the results using genus

level means of longevity and climate. It confirms again that longevity-climate relationships remain robust within both groups (Supplementary Fig. 7a,b).

Table 2. Standard and phylogenetically controlled correlations for the key variables affecting variation in longevity (cf. Supplementary Table 1). Standard correlations are Pearson’s correlations and phylogenetic controlled correlations were performed using phylogenetic generalised linear models (PGLS) (see methods).

	Angiosperms						Gymnosperms					
	Standard correlation			Phylogenetically controlled			Standard correlation			Phylogenetically controlled		
	R	p value	n	R	p value	n	R	p value	n	R	p value	n
Mean radial growth*	-0.564	0.000	429	-0.507	0.000	381	-0.567	0.000	184	-0.467	0.000	179
Mean annual temperature	0.030	0.488	534	0.038	0.405	476	-0.378	0.000	205	-0.420	0.000	200
Precipitation driest quarter**	0.332	0.000	488	0.277	0.000	432	0.080	0.258	203	0.072	0.312	198
Proximity to tree line	-0.071	0.108	519	0.048	0.302	462	0.397	0.000	201	0.429	0.000	196
Potential Net Primary Productivity	0.115	0.009	517	0.075	0.108	460	-0.359	0.000	201	0.423	0.000	196
Wood density	0.124	0.009	441	0.091	0.069	399	-0.081	0.333	144	0.123	0.146	140
Maximum tree height	0.351	0.000	325	0.294	0.000	291	0.200	0.005	198	0.261	0.000	193
Leaf Mass per Area*	-0.048	0.599	124	0.082	0.392	109	0.175	0.287	39	0.282	0.082	38
Nitrogen per Mass	-0.284	0.001	124	0.220	0.020	109	0.009	0.957	40	0.002	0.991	39

* These data were log₁₀-transformed for the analysis; ** Correlations with precipitation and Climate moisture index exclude species that have their main occurrence in floodplains or wetlands.

Supplementary Figure 7. Relationships between longevity and the two most important climate, variables at genus level. For each genus we calculated the mean longevity and mean temperature (a) and precipitation during the driest quarter (b). These results show that the key patterns are not driven simply by few widespread genera showing strong relationships with climate but hold up at genus level.

3. Climate change. The authors responded that “We realised this section was

outside the scope of the article, and it was also criticised by Reviewer 1. We have now deleted this entire section". But in the introduction, the following sentence remains: "These insights can support predictions of future responses to global change, for example, by providing Earth System Models with more realistic estimates of tree turnover" (Line 96-98). However, the climate and longevity relationships are mainly correlations, as shown in all the three figures in the main text. Do you really believe that your correlations between climate and longevity can contribute to Earth System Models with more realistic estimates of tree turnover?

>> This seems a minor comment, but we do believe that empirical estimates of tree turnover rates and their variation across biomes (as we show in our manuscript, see Fig, 1b) can help Earth System models. This is what the modelling community themselves clearly state. For example, the study by Friend et al. (2014) that we cite, states that "*Further work should focus on confronting the processes and emergent model behavior responsible for these discontinuities with observational data. Spatial and temporal variability in carbon residence time and tree mortality is an obvious place to start, and although recent studies have identified important sources of relevant data (e.g., refs. 20 and 21), data on key processes such as mortality at large scales are rare.*"

We believe it is not such a stretch to state this, and thus prefer to keep it in the manuscript.

4. Conclusion. The authors responded that I had misunderstood their conclusion and emphasized in their responses that "when pooling the angiosperms and gymnosperms most traits were significantly related to longevity (first left columns), but these relationships disappeared, or became weaker, when the analysis was done per group (columns to the right). We go from 12 significant trait longevity relationships (when the two groups are pooled) to much fewer relationships for the groups separately (i.e., 4 for Angiosperms and 2 for conifers)". Do you mean "pooling the angiosperms and gymnosperms" is better than separating them in explaining the role of traits? If so, why did you separate them?

>> We mean that when pooling all data of angiosperms and gymnosperms we do find that all traits (except for one) show significant correlations with longevity. One could indeed thus conclude that traits are better in explaining longevity variation when data are pooled.

Why did we separate them? We separated them firstly because we know from literature that the two groups have very different traits, growth strategies and longevities, affecting the observed overall relationship (ie when both groups are pooled). Secondly, we wanted to avoid taxonomic biases due to the long evolutionary differences between groups and their difference in biogeographic distribution. We make this clear at the end of the introduction where we say "*Previous studies showed that angiosperms and gymnosperms differ strongly in their growth strategies⁷, photosynthetic capacity⁸, hydraulic traits⁹⁻¹¹ and longevity^{1,12,13}, which we confirm in our analysis of the full dataset. Given that these taxonomic groups diverged over 300 million years ago¹⁴ and that they differ in their global distribution¹⁵, we primarily analyse the results for these groups separately to avoid any phylogenetic bias in the results.*"

- 1 Piovesan, G. & Biondi, F. On tree longevity. *New Phytologist* **231**, 1318-1337 (2021).
- 2 Smith, S. A. & Brown, J. W. Constructing a broadly inclusive seed plant phylogeny. *Am. J. Bot.* **105**, 302-314 (2018).
- 3 Orme, D. *et al.* The caper package: comparative analysis of phylogenetics and evolution in R. *R package version* **5**, 1-36 (2013).
- 4 Revell, L. J. phytools 2.0: an updated R ecosystem for phylogenetic comparative methods (and other things). *PeerJ* **12**, e16505 (2024).
- 5 Paradis, E. & Schliep, K. ape 5.0: an environment for modern phylogenetics and evolutionary analyses in R. *Bioinformatics* **35**, 526-528 (2019).
- 6 Bates, D., Mächler, M., Bolker, B. & Walker, S. Fitting Linear Mixed-Effects Models Using lme4. *2015* **67**, 48 (2015). <https://doi.org/10.18637/jss.v067.i01>
- 7 Bond, W. The tortoise and the hare: ecology of angiosperm dominance and gymnosperm persistence. *Biol. J. Linn. Soc.* **36**, 227-249 (1989).
- 8 Lusk, C. H., Wright, I. & Reich, P. B. Photosynthetic differences contribute to competitive advantage of evergreen angiosperm trees over evergreen conifers in productive habitats. *New Phytol.* **160**, 329-336 (2003).
- 9 Brodribb, T. J. & Feild, T. S. Leaf hydraulic evolution led a surge in leaf photosynthetic capacity during early angiosperm diversification. *Ecol. Lett.* **13**, 175-183 (2010). <https://doi.org/10.1111/j.1461-0248.2009.01410.x>
- 10 Johnson, D. M., McCulloh, K. A., Woodruff, D. R. & Meinzer, F. C. Hydraulic safety margins and embolism reversal in stems and leaves: why are conifers and angiosperms so different? *Plant Sci.* **195**, 48-53 (2012).
- 11 Sperry, J. S., Hacke, U. G. & Pittermann, J. Size and function in conifer tracheids and angiosperm vessels. *Am. J. Bot.* **93**, 1490-1500 (2006).
- 12 Liu, J. *et al.* Age and spatial distribution of the world's oldest trees. *Conserv. Biol.* **36**, e13907 (2022).
- 13 Locosselli, G. M. *et al.* Global tree-ring analysis reveals rapid decrease in tropical tree longevity with temperature. *Proceedings of the National Academy of Sciences* **117**, 33358-33364 (2020).
- 14 Willis, K. J. & McElwain, J. C. *The evolution of plants*. (Oxford University Press, USA, 2014).
- 15 Ma, H. *et al.* The global biogeography of tree leaf form and habit. *Nature plants*, 1-15 (2023).

Reviewer #3 (Remarks to the Author):

This manuscript addresses important questions related to variation in tree longevity and its ecological drivers. I recognize that this work has already undergone prior rounds of peer review, which have helped resolve many technical details and improve clarity. The authors' responses demonstrate significant effort to strengthen the manuscript, and I agree that the study presents novel insights, particularly the identification of two contrasting pathways to tree longevity across environmental gradients.

Nevertheless, I believe some conceptual uncertainties remain, which are inherently difficult to resolve fully but are worth considering. In addition, I would like to raise another of these difficult-to-address issues, which could have implications for the interpretation of the results. Given that gymnosperms tend to be more prevalent in environments subjected to freezing temperatures, I wonder whether similar relationships between temperature, precipitation, and longevity would emerge if gymnosperms and angiosperms were pooled and analyzed collectively, but separately for environments with and without freezing conditions. Such an approach could help disentangle whether the observed patterns are primarily driven by phylogenetic differences, environmental filtering, or their interaction.

In summary, I believe the manuscript, as it stands, contributes valuable and novel perspectives to ongoing research on variation in tree longevity.

We thank the reviewer for the time taken to read the manuscript.

On the suggestion to do a pooled analysis, we would like to note that we have been cautious with the interpretation of any pooled analysis, because of the large intrinsic phylogenetic differences between the two groups. This was also remarked upon by both previous reviewers. We now completed the suggested analysis to pool the samples.

For the pooled species in freezing environments, defined as any species with a minimum temperature in its range lower than 0 degrees, we find patterns that are very similar to those of the gymnosperms in general (see figure below). This is not surprising given that only 19 Gymnosperm species occur in entirely frost-free environments out of the 205 Gymnosperms in our dataset.

Similarly, the analysis of frost-free species, shows large similarities to the results of the Angiosperms, because the majority of the Angiosperms occurs in warm (frost free) environments (315 of the 534).

The only differences we find compared to the original results arise from pooling of the two phylogenetically different groups with very different leaf traits, resulting in stronger effects of LMA and nitrogen per mass (similar to the overall pooled dataset, cf. Supplementary Table 1 – first column). One interesting new result is that the association of WD with longevity is opposite in freezing vs. frost free conditions (see subpanel e). It is known that WD increases with an increase in temperature, which is likely to explain the negative relationship for species restricted to environments with freezing temperatures.

While interesting, we feel that the suggested separation of species into frost-free vs. frost environments did not help much in disentangling phylogeny and environment, and it still

leaves open the question to what degree observed differences are due to environmental filtering vs. intrinsic phylogeny effects. We have now added a section to the discussion to reflect this uncertainty:

Lines 354-360 “Finally, differences in the climate response may also emerge from environmental filtering for different climates, as gymnosperms occur predominantly in environments that experience freezing during some parts of the year. In line with this we find that separation of the dataset into species occurring in freezing versus non-freezing environments results in similar relationships with temperature and precipitation as those observed for gymnosperms and angiosperms (Supplementary Fig. 9). These results show that more research is needed to assess interactions between environment and phylogeny, and to fully disentangle the relative roles of these various mechanisms in controlling tree survival and explain differences in associations between longevity and climate.”

Supplementary Figure 9. Relationships between longevity and all variables shown in main Figure 2 with species separated into those experiencing frost (n= 405) versus species without frost (n= 334). Note that we here combined the entire dataset, irrespective of taxonomic group (angiosperms and gymnosperms). Also note that nearly all gymnosperms (186 out of 205 species) fall into the category of species experiencing frost, and that the category of species with frost mainly consists of angiosperms.

Reviewer #4

This was my first review of this manuscript. Overall, this was a very interesting manuscript that took a thoughtful approach to pursuing an important question. I found the responses to prior review both thorough and compelling, and I have no additional comments related to those critiques. I look forward to citing this study.

To inform my reading of these responses to review, I also thoroughly read the manuscript (which I quite enjoyed). My instructions were to avoid providing a full-review of the manuscript (which I did not), but my reading of the text raised two major concerns. I realize that this manuscript has likely undergone intensive prior review and my role here was not to provide an in-depth review of the complete manuscript, but I am sharing these concerns because (1) I think they are scientifically important and (2) if these concerns are not addressed, then readers like myself will be (hopefully unnecessarily) skeptical of the published results. I expect that it will be straightforward to address these concerns, particularly major comment #1, and addressing them will produce a product that is a more durable contribution to the literature. I also highlight a couple very minor items that could be addressed.

We thank the reviewer for the time taken and the suggestions to which we respond below.

Major comments

1. Claims of causation do not appear to be supported (Lines 80-82, 202-216, 302-303, Fig. 3, and elsewhere). There are two major logical flaws in the conclusions listed in the abstract, which appear to exist because of how structural equation modeling was applied and interpreted. Fundamentally, the study does not justify why tree height and growth mechanistically cause differences in longevity, and I am not aware of data that would support a direct causal relationship between these variables. I break down these issues around the two conclusions listed in the abstract, but they apply throughout the text. These claims either need to be justified with data or the text should be reframed to highlight that these are statistical associations without directly causative effects.

The first unsupported claim in the abstract (line 80: "*First, higher water availability increases maximum tree height allowing species to attain greater longevity*") is circular and it is missing a logical link. Trees could just as reasonably be taller because they live longer, but the text claims that trees live longer because they are taller. Indeed, our metrics of maximum tree height are based on our field observations, which are strongly sensitive to sample size; this is problematic because the number and maximum height of very tall trees observed within a species will increase with their longevity (due to survivor bias and continued growth with survival), leading to greater observed

height with greater longevity. Additionally, there is no justification for why trees being taller would cause them to be older. In simplistic terms, it is well known that taller individuals can be much younger than shorter individuals. There must be an additional conceptual link for why taller trees mechanistically live longer for this claim to be supported.

The second claim highlighter in the abstract (line 81: “:Secondly greater water availability increases stand density and inter-tree competition, limiting growth thereby increasing tree lifespan”) asserts that higher growth directly causes lower survivorship. While high growth is correlated with higher mortality, I am not aware of any conceptual or empirical work demonstrating a direct mechanistic effect of growth on mortality. Fundamentally, trees do not die because they were productive and increased in diameter. High growth may be correlated with tradeoffs against other traits that could cause susceptibility to an agent of mortality, but that is not the same as higher growth causing low survivorship. The study provides some speculation about these connections on lines 306-316, but this text is speculative and key empirical connections are untested or uncertain (e.g., the fact that some shade-tolerant trees can survive for a long time with low growth in shaded conditions doesn’t mean that slow growth caused them to live a long time).

The strong and empirically supported results in this study will carry more weight and inspire more overall confidence in this study if these unsupported claims of causation were supported by empirical data or properly conveyed as speculation.

We agree these issues require more careful phrasing, and we thus reformulated the following text to reflect a more cautious interpretation of the findings (changed wording is underlined):

Abstract, Lines 80-83: “First, higher water availability increases species’ maximum tree height which is associated with greater longevity. Secondly, greater water availability increases stand density and inter-tree competition, limiting growth which may increase tree lifespan.

In the discussion (lines 257-258) instead of referring to mechanisms, we now say we provide “insights in the ecological factors contributing to greater longevity for gymnosperms in cold climates (i.e., low growth) and for angiosperms in humid climates (i.e., increased tree height and competition),”

Line 309, on the effects of tree height : “Increases in precipitation lead to greater maximum tree height¹⁻⁴, which in turn is associated with tree longevity.”

Line 322, on effects of slow growth in understory: “These may represent important ecological processes that contribute to extending the longevity of shade-tolerant species.”

Final Summary paragraph, Line 460: “In this productivity pathway increases in competition limits individual growth for trees in the understory and allows others to reach large statures, both of which are associated with greater longevity. “

2. I am concerned about the accuracy and reliability of the demography-based estimates of longevity. I have low confidence in the validity of the growth-based or mortality-based estimates of tree longevity (the methods text focuses on growth, but Table S3 says growth and mortality rates), particularly in tropical forests where this method appears to have been applied to a large proportion of species. Fundamentally, we know (1) that growth and mortality rates change with tree size, and (2) our data on species-level growth and mortality rates is very limited for giant trees and, for most species, the imprecision of data for giant individuals is so high that they are unreliable. My concern is that if we know we lack the data to properly characterize the growth and survivorship of giant trees (which we know will strongly influence

longevity), then we should suspect that we cannot accurately estimate the maximum potential longevity of these species. These problems will be exaggerated in tropical forests where diversity is so high and sampling effort is so low that we lack sufficient species-specific demographic data to be able to capture long-term trends.

To highlight this issue, I share some soon-to-be-published data comparing mortality rates from the only study site in the tropics with enough species-specific data (ca. 9 million years of tree-level monitoring within a single community) to estimate mortality rates for many giant tree taxa (N = 37 species in this case). These data show a general positive correlation between species-specific mortality rates as giant trees (y-axis: >50 cm in diameter in this case; points are species-level means with 95% confidence intervals) versus species-specific mortality rates for all trees <1cm in diameter within the same species (x-axis). However, for very long-lived taxa (annual mortality rates below 1.0 % year as giant trees, suggesting mean survivorship of >100 years after reaching 50cm DBH), there is no predictive relationship between their mortality as giant trees and their population-wide mortality rates. Similarly, species with the lowest population-wide mortality rates (~1%) exhibited >5-fold variation in their mortality rates as giant trees, meaning that true maximum longevity should differ enormously even among taxa with the lowest population-level mortality rates. If growth exhibits the same general pattern as mortality, then we might expect growth-based predictions of longevity for these longest-lived canopy tree species (and therefore for angiosperms in wet, warm, and productive locations) to chronically underestimate their longevity. Overall, these patterns are concerning because – if the growth-based estimates of longevity are biased - then any climatic, phylogenetic (angiosperm versus gymnosperm), or geographic biases in the distribution of growth-based estimates of longevity could thereby influence associations with climate, phylogeny, and/or geography.

I see three potential options to addressing this issue, although presumably there are alternative solutions as well. First, this study does not provide details of exactly how the growth-based projections of longevity were performed or what steps were taken to ensure that they were robust to the potential issues described above. If appropriate steps were taken to ensure that these estimates were robust, then the authors should add text detailing these reasons (I expect that there simply are not enough data on size-dependent mortality and growth to properly test these trends, particularly in tropical regions).

Second, the authors could take the omission route they already employed for evaluating whether

their results were robust to the inclusion of the lowest confidence longevity data. Specifically, the authors could re-run their analyses only including confidence levels 3 and 4 and evaluate the geographic coverage of this reduced dataset. This would tell us whether the results of the study depend on our confidence in the extrapolation of longevity from tree demography. Third, the authors could re-run the models on simulated data that assume either bias or imprecision to the growth-based projections of longevity (for example, re-run the models 1,500 times with 500 iterations adding random noise to the growth-based longevity data, 500 adding random positive bias to these data, and 500 adding random negative bias to these data), thereby quantifying how robust the results are to imprecision and bias in these lower-confidence estimates. If the authors find that their original results are not always supported, they could still draw similar conclusions, but with clear caveats that these conclusions require confidence in extrapolations based on limited data.

We share these concerns and are fully aware of this shortcoming as well as several other possible biases (see the online methods). As these growth projections are based on published datasets, solutions 1 and 3 are very difficult to implement. But we did check whether simply leaving these estimates based on (uncertain) growth projections out affected the results (solution 2), and find that the results remained robust. We have added this set of figures as another supplementary Figure 4, and copied it below.

We further added the following text to the online methods:

Lines : 525-530 : *“These age estimates based on projections using short-term growth and mortality rates are subject to several potential biases due to limited data availability, variation in growth and mortality with tree size, and trade-offs between growth and mortality within species. However, our results remained robust even when excluding growth projections from our analysis (Supplementary Fig. 4).”*

Supplementary Figure 4. Relationships between species' longevity and most important climate, environmental and trait variables excluding longevity data based on growth projections (see methods and ED Table 3c). Variables include species' mean annual temperature (a) and precipitation during the driest quarter (b), species' mean proximity to the estimated tree line position (c), potential net primary productivity (d), and species' functional traits including wood density (e), maximum species tree height (f), and leaf mass per area (g) and leaf nitrogen (h). Trend lines are fitted using a linear model. Statistics show the correlation coefficient (R) and significance levels (p) for the linear relationships. Data were grouped by the two major taxonomic groups in the dataset, angiosperms and gymnosperms. Note that data for Leaf Mass per Area were \log_{10} transformed.

Minor comments

Lines 77-78 (here and elsewhere). This study only provides information on a tiny fraction of angiosperms, with biased sampling in terms of both methods and geography. Consider revising the conclusions to be a bit more cautious and representative of the data. Specifically, we do not know how ‘most long-lived angiosperms’ associate with any factor because we have not yet identified most of the long-lived angiosperms in nature. It would be more accurate to discuss tendencies or associations of angiosperms, rather than speaking of them as a monolithic unit.

We agree and have now changed the wording to “long-lived angiosperms tend to follow ...” (and used similar wording for gymnosperms).

Lines 112-114. This statement overly simplifies the literature to the degree that it is misleading. Specifically, it implies that variation in longevity is largely associated with the slow-fast continuum. However, the slow-fast continuum is almost entirely based on demographic differences among saplings and subcanopy trees, whereas many long-lived taxa ultimately persist for centuries in their local canopy. It is well-established that additional axes of demographic variation (e.g., the stature-recruitment axis) also play a major role in structuring community composition, assembly, and demography. Importantly, additional axes like the stature-recruitment tradeoff are orthogonal to the slow-fast continuum and this orthogonal axis is driven by demographic differences associated with tree size, particularly large and long-lived individuals that can fall at many locations along the slow-fast continuum (Ruger et al. 2018, 2020). It would be more accurate to revise this text to emphasize that the slow-fast continuum is one of multiple key axes of demographic variation that likely associated with longevity.

Rüger, N., L. S. Comita, R. Condit, D. Purves, B. Rosenbaum, M. D. Visser, S. Joseph Wright, and C. Wirth. 2018. Beyond the fast–slow continuum: demographic dimensions structuring a tropical tree community. *21* 7:1075–1084.

Rüger, N., R. Condit, D. H. Dent, S. J. DeWalt, S. P. Hubbell, J. W. Lichstein, O. R. Lopez, C. Wirth, and C. E. Farris. 2020. Demographic trade-offs predict tropical forest dynamics. *Science* 368:165.

We agree and now discuss this independent axis of “stature-recruitment” explicitly in our Discussion (see lines 397 and following Discussion). To avoid this simplification in the Introduction, we rephrased the text as follows:

These differences arise from the diversification of species along different axes of demographic variation. One of these axes is the well-established fast-slow continuum of life-history strategies shaped by trade-offs between growth and survival⁵.

Line 320-321. The claim that there is global convergence in hydraulic safety margins should be omitted because this hypothesis was unsupported once confronted with more detailed sampling in the tropics. Specifically, better data from tropical forests has shown that hydraulic safety margins are highly variable among sites, refuting the hypothesis that there is global convergence in hydraulic safety margins.

Tavares, J. V., et al. 2023. Basin-wide variation in tree hydraulic safety margins predicts the carbon balance of Amazon forests. *Nature* 617:111–117.

We believe this is still open for debate, but now added this publication and Sanchez-Martinez et al. (2023), showing that hydraulic safety margins may vary between sites:

Lines 327-329: “This may point to a convergence of species’ hydraulic safety margins across the globe⁶, although other studies report that variation in hydraulic safety margins are a good predictor for drought related mortality^{7,8}. More data on hydraulic properties are needed to fully explore these effects for a greater number of species.”

Line 506. Significantly should not have the ‘ly’
We changed this.

Lines 532-534. Wood density may be the best proxy available, but that does not suggest that wood density is actually a good proxy for the species-level life history strategies that determine longevity. Wood density is a decent proxy for the slow-fast continuum, which is a tradeoff based on trends among understory trees. However, recent data from tropical forest demography has shown that tree survivorship as large individuals is related to a demographic axis (‘stature-recruitment’) that is orthogonal to the slow-fast continuum and largely unrelated to wood density (see the Ruger papers cited above). It is very likely that this orthogonal axis is more relevant to maximum tree age than the slow-fast continuum and wood density. The wood density-based result should give you confidence that you have sampled a single axis of life history strategy (one that is largely associated with small-tree performance), but it is not evidence that your sampling is representative of life history strategies that control tree longevity.

We are aware of this and initially showed only wood density as it was the best proxy available. Databases for other traits such the tree height database (Tallo) may contain biases wrt species’ maximum tree height as it depends on the sampling effort for each species. For completeness, we have now chosen to include this comparison in the manuscript. When restricting this reference height dataset to those species with at least 50 trees, we observe a relatively close match of tree height with our dataset. However, we acknowledge that for Angiosperms the reference dataset has a larger share of small species.

We added the following text to the online methods of the ms : “*This comparison included wood density as it is closely linked to life history variation along the fast-slow axis and other functional attributes of trees⁹⁻¹¹ as well as tree height as a proxy for plant stature which is the second most important functional traits axis in trees¹²⁻¹⁴. This shows that both the average and range of wood density of our study are very similar to that of the globe for both angiosperm and gymnosperms (Supplementary Fig. 2a,b). It further shows very close agreement with tree height, although the functional trait dataset has a somewhat greater share of short statured trees for angiosperms (Supplementary Fig., 2c,d).* “

Supplementary Figure 2. Comparison of wood density (a,b) and maximum tree height (c,d) of the sampled species in this study to that of global functional trait datasets. For wood density we used the global wood density database¹⁵ and for maximum tree height we used *tallo*, a tree allometry database¹⁶. The vertical lines indicate the means of the two datasets. The mean wood density (WD) of our study (angiosperms = 0.61 g/cm³, gymnosperms = 0.45 g/cm³) compares well to that from the global wood density database (angiosperms = 0.62 g/cm³, 8,150 species, gymnosperms = 0.45 g/cm³, 262 species) and to the community weighted wood density from reference¹⁷ (angiosperms = 0.59 g/cm³, 8,036 species; gymnosperms = 0.47 g/cm³, 213 species). The maximum tree height of our study (angiosperms = 30.5 m, gymnosperms = 36.2 m) compares well to that from the *tallo* database¹⁶ when restricted to those species with 50 records or more (angiosperms = 26.4 m, 992 species, gymnosperms = 36.5 m, 65 species).

- 1 Scheffer, M. *et al.* A global climate niche for giant trees. *Global Change Biol.* **24**, 2875-2883 (2018).
- 2 Moles, A. T. *et al.* Global patterns in plant height. *J. Ecol.* **97**, 923-932 (2009).
- 3 Liu, H. *et al.* Hydraulic traits are coordinated with maximum plant height at the global scale. *Science Advances* **5**, eaav1332 (2019).
- 4 Klein, T., Randin, C. & Körner, C. Water availability predicts forest canopy height at the global scale. *Ecol. Lett.* **18**, 1311-1320 (2015).
- 5 Stearns, S. C. Trade-offs in life-history evolution. *Funct. Ecol.* **3**, 259-268 (1989).
- 6 Choat, B. *et al.* Global convergence in the vulnerability of forests to drought. *Nature* **491**, 752 (2012).
- 7 Sanchez-Martinez, P. *et al.* Increased hydraulic risk in assemblages of woody plant species predicts spatial patterns of drought-induced mortality. *Nature Ecology & Evolution* **7**, 1620-1632 (2023).
- 8 Tavares, J. V. *et al.* Basin-wide variation in tree hydraulic safety margins predicts the carbon balance of Amazon forests. *Nature* **617**, 111-117 (2023).
- 9 Chave, J. *et al.* Towards a worldwide wood economics spectrum. *Ecol. Lett.* **12**, 351-366 (2009). <https://doi.org/10.1111/j.1461-0248.2009.01285.x>
- 10 Weedon, J. T. *et al.* Global meta-analysis of wood decomposition rates: a role for trait variation among tree species? *Ecol. Lett.* **12**, 45-56 (2009). <https://doi.org/10.1111/j.1461-0248.2008.01259.x>
- 11 Hacke, U. G., Sperry, J. S., Pockman, W. T., Davis, S. D. & McCulloh, K. A. Trends in wood density and structure are linked to prevention of xylem implosion by negative pressure. *Oecologia* **126**, 457-461 (2001). <https://doi.org/10.1007/s004420100628>
- 12 Maynard, D. S. *et al.* Global relationships in tree functional traits. *Nature Communications* **13**, 3185 (2022).
- 13 Joswig, J. S. *et al.* Climatic and soil factors explain the two-dimensional spectrum of global plant trait variation. *Nature Ecology & Evolution* **6**, 36-50 (2022).
- 14 Diaz, S. *et al.* The global spectrum of plant form and function. *Nature* **529**, 167-171 (2016). <https://doi.org/10.1038/nature16489>
- 15 Zanne, A. E. *et al.* Global wood density database. *Dryad. Identifier:* <http://hdl.handle.net/10255/dryad.235>. (2009).
- 16 Jucker, T. *et al.* Tallo: A global tree allometry and crown architecture database. *Global Change Biol.* **28**, 5254-5268 (2022).
- 17 Mo, L. *et al.* The global distribution and drivers of wood density and their impact on forest carbon stocks. *Nature Ecology & Evolution*, 1-18 (2024).

Rebuttal letter

REVIEWERS' COMMENTS

Reviewer #3 (Remarks to the Author):

I greatly appreciate the additional analysis separating species into freezing versus frost-free environments, along with its thoughtful integration into the discussion. Importantly, the authors acknowledge that uncertainties remain regarding the extent to which observed differences are driven by environmental filtering versus intrinsic phylogenetic effects, which adds valuable nuance and depth to the manuscript.

I am fully satisfied with the authors' revisions and responses. I congratulate them on this significant contribution and look forward to the publication of the article, which will be an important resource to cite.

We thank the reviewer for their comments.

Reviewer #4 (Remarks to the Author):

This is a strong paper, and the final round of revisions was generally thorough and appropriate. I only have one final minor comment. The authors stated that their results were robust to the exclusion of the growth-based and mortality-based projections of tree longevity, but omitting these data actually caused the association between wood density and Angiosperm longevity to become non-significant (Supplementary Fig. 4e). This should be stated explicitly in the results (lines 200-201) so that it is clear for the reader that this association does not hold up when those data are excluded.

We thank the reviewer for pointing out this oversight and we have now stated this explicitly in the manuscript in the results section by adding "*Our results remained largely robust even when excluding growth projections from our analysis although the relationship with wood density for angiosperms disappeared (Supplementary Fig. 4).*" And adding a comment to the methods section where the reason for the exclusion of growth projections is explained.

This was my first review of this manuscript. Overall, this was a very interesting manuscript that took a thoughtful approach to pursuing an important question. I found the responses to prior review both thorough and compelling, and I have no additional comments related to those critiques. I look forward to citing this study.

To inform my reading of these responses to review, I also thoroughly read the manuscript (which I quite enjoyed). My instructions were to avoid providing a full-review of the manuscript (which I did not), but my reading of the text raised two major concerns. I realize that this manuscript has likely undergone intensive prior review and my role here was not to provide an in-depth review of the complete manuscript, but I am sharing these concerns because (1) I think they are scientifically important and (2) if these concerns are not addressed, then readers like myself will be (hopefully unnecessarily) skeptical of the published results. I expect that it will be straightforward to address these concerns, particularly major comment #1, and addressing them will produce a product that is a more durable contribution to the literature. I also highlight a couple very minor items that could be addressed.

Major comments

1. Claims of causation do not appear to be supported (Lines 80-82, 202-216, 302-303, Fig. 3, and elsewhere). There are two major logical flaws in the conclusions listed in the abstract, which appear to exist because of how structural equation modeling was applied and interpreted. Fundamentally, the study does not justify why tree height and growth mechanistically cause differences in longevity, and I am not aware of data that would support a direct causal relationship between these variables. I break down these issues around the two conclusions listed in the abstract, but they apply throughout the text. These claims either need to be justified with data or the text should be reframed to highlight that these are statistical associations without directly causative effects.

The first unsupported claim in the abstract (line 80: *“First, higher water availability increases maximum tree height allowing species to attain greater longevity”*) is circular and it is missing a logical link. Trees could just as reasonably be taller because they live longer, but the text claims that trees live longer because they are taller. Indeed, our metrics of maximum tree height are based on our field observations, which are strongly sensitive to sample size; this is problematic because the number and maximum height of very tall trees observed within a species will increase with their longevity (due to survivor bias and continued growth with survival), leading to greater observed height with greater longevity. Additionally, there is no justification for why trees being taller would cause them to be older. In simplistic terms, it is well known that taller individuals can be much younger than shorter individuals. There must be an additional conceptual link for why taller trees mechanistically live longer for this claim to be supported.

The second claim highlighted in the abstract (line 81: *“Secondly, greater water availability increases stand density and inter-tree competition, limiting growth thereby increasing tree lifespan.”*) asserts that

higher growth directly causes lower survivorship. While high growth is correlated with higher mortality, I am not aware of any conceptual or empirical work demonstrating a direct mechanistic effect of growth on mortality. Fundamentally, trees do not die because they were productive and increased in diameter. High growth may be correlated with tradeoffs against other traits that could cause susceptibility to an agent of mortality, but that is not the same as higher growth causing low survivorship. The study provides some speculation about these connections on lines 306-316, but this text is speculative and key empirical connections are untested or uncertain (e.g., the fact that some shade-tolerant trees can survive for a long time with low growth in shaded conditions doesn't mean that slow growth caused them to live a long time).

The strong and empirically supported results in this study will carry more weight and inspire more overall confidence in this study if these unsupported claims of causation were supported by empirical data or properly conveyed as speculation.

2. I am concerned about the accuracy and reliability of the demography-based estimates of longevity.

I have low confidence in the validity of the growth-based or mortality-based estimates of tree longevity (the methods text focuses on growth, but Table S3 says growth and mortality rates), particularly in tropical forests where this method appears to have been applied to a large proportion of species. Fundamentally, we know (1) that growth and mortality rates change with tree size, and (2) our data on species-level growth and mortality rates is very limited for giant trees and, for most species, the imprecision of data for giant individuals is so high that they are unreliable. My concern is that if we know we lack the data to properly characterize the growth and survivorship of giant trees (which we know will strongly influence longevity), then we should suspect that we cannot accurately estimate the maximum potential longevity of these species. These problems will be exaggerated in tropical forests where diversity is so high and sampling effort is so low that we lack sufficient species-specific demographic data to be able to capture long-term trends.

To highlight this issue, I share some soon-to-be-published data comparing mortality rates from the only study site in the tropics with enough species-specific data (ca. 9 million years of tree-level monitoring within a single community) to estimate mortality rates for many giant tree taxa (N = 37 species in this case). These data show a general positive correlation between species-specific mortality rates as giant trees (y-axis: >50 cm in diameter in this case; points are species-level means with 95% confidence intervals) versus species-specific mortality rates for all trees <1cm in diameter within the same species (x-axis). However, for very long-lived taxa (annual mortality rates below 1.0 % year as giant trees, suggesting mean survivorship of >100 years after reaching 50cm DBH), there is no predictive relationship between their mortality as giant trees and their population-wide mortality rates. Similarly, species with the lowest population-wide mortality rates (~1%) exhibited >5-fold variation in their mortality rates as giant trees, meaning that true maximum longevity should differ enormously even among taxa with the lowest population-level mortality rates. If growth exhibits the same general pattern as mortality, then we might expect growth-based predictions of longevity for these longest-lived canopy tree species (and therefore for angiosperms in wet, warm, and productive locations) to chronically

underestimate their longevity. Overall, these patterns are concerning because – if the growth-based estimates of longevity are biased - then any climatic, phylogenetic (angiosperm versus gymnosperm), or geographic biases in the distribution of growth-based estimates of longevity could thereby influence associations with climate, phylogeny, and/or geography.

I see three potential options to addressing this issue, although presumably there are alternative solutions as well. First, this study does not provide details of exactly how the growth-based projections of longevity were performed or what steps were taken to ensure that they were robust to the potential issues described above. If appropriate steps were taken to ensure that these estimates were robust, then the authors should add text detailing these reasons (I expect that there simply are not enough data on size-dependent mortality and growth to properly test these trends, particularly in tropical regions). Second, the authors could take the omission route they already employed for evaluating whether their results were robust to the inclusion of the lowest confidence longevity data. Specifically, the authors could re-run their analyses only including confidence levels 3 and 4 and evaluate the geographic coverage of this reduced dataset. This would tell us whether the results of the study depend on our confidence in the extrapolation of longevity from tree demography. Third, the authors could re-run the models on simulated data that assume either bias or imprecision to the growth-based projections of longevity (for example, re-run the models 1,500 times with 500 iterations adding random noise to the growth-based longevity data, 500 adding random positive bias to these data, and 500 adding random negative bias to these data), thereby quantifying how robust the results are to imprecision and bias in these lower-confidence estimates. If the authors find that their original results are not always supported, they could still draw similar conclusions, but with clear caveats that these conclusions require confidence in extrapolations based on limited data.

Minor comments

Lines 77-78 (here and elsewhere). This study only provides information on a tiny fraction of angiosperms, with biased sampling in terms of both methods and geography. Consider revising the conclusions to be a bit more cautious and representative of the data. Specifically, we do not know how

‘most long-lived angiosperms’ associate with any factor because we have not yet identified most of the long-lived angiosperms in nature. It would be more accurate to discuss tendencies or associations of angiosperms, rather than speaking of them as a monolithic unit.

Lines 112-114. This statement overly simplifies the literature to the degree that it is misleading. Specifically, it implies that variation in longevity is largely associated with the slow-fast continuum. However, the slow-fast continuum is almost entirely based on demographic differences among saplings and subcanopy trees, whereas many long-lived taxa ultimately persist for centuries in their local canopy. It is well-established that additional axes of demographic variation (e.g., the stature-recruitment axis) also play a major role in structuring community composition, assembly, and demography. Importantly, additional axes like the stature-recruitment tradeoff are orthogonal to the slow-fast continuum and this orthogonal axis is driven by demographic differences associated with tree size, particularly large and long-lived individuals that can fall at many locations along the slow-fast continuum (Ruger et al. 2018, 2020). It would be more accurate to revise this text to emphasize that the slow-fast continuum is one of multiple key axes of demographic variation that likely associated with longevity.

Rüger, N., L. S. Comita, R. Condit, D. Purves, B. Rosenbaum, M. D. Visser, S. Joseph Wright, and C. Wirth. 2018. Beyond the fast–slow continuum: demographic dimensions structuring a tropical tree community. *21* 7:1075–1084.

Rüger, N., R. Condit, D. H. Dent, S. J. DeWalt, S. P. Hubbell, J. W. Lichstein, O. R. Lopez, C. Wirth, and C. E. Farrior. 2020. Demographic trade-offs predict tropical forest dynamics. *Science* 368:165.

Line 320-321. The claim that there is global convergence in hydraulic safety margins should be omitted because this hypothesis was unsupported once confronted with more detailed sampling in the tropics. Specifically, better data from tropical forests has shown that hydraulic safety margins are highly variable among sites, refuting the hypothesis that there is global convergence in hydraulic safety margins.

Tavares, J. V., et al. 2023. Basin-wide variation in tree hydraulic safety margins predicts the carbon balance of Amazon forests. *Nature* 617:111–117.

Line 506. Significantly should not have the ‘ly’

Lines 532-534. Wood density may be the best proxy available, but that does not suggest that wood density is actually a good proxy for the species-level life history strategies that determine longevity. Wood density is a decent proxy for the slow-fast continuum, which is a tradeoff based on trends among understory trees. However, recent data from tropical forest demography has shown that tree survivorship as large individuals is related to a demographic axis (‘stature-recruitment’) that is orthogonal to the slow-fast continuum and largely unrelated to wood density (see the Ruger papers cited above). It is very likely that this orthogonal axis is more relevant to maximum tree age than the

slow-fast continuum and wood density. The wood density-based result should give you confidence that you have sampled a single axis of life history strategy (one that is largely associated with small-tree performance), but it is not evidence that your sampling is representative of life history strategies that control tree longevity.